# Health system interventions to integrate genetic testing in routine oncology services: A systematic review

**Rosie O'Shea**[1,2]*, **Natalie Taylor**[1,3‡], **Ashley Crook**[2°], **Chris Jacobs**[2°], **Yoon Jung Kang**[1,3‡], **Sarah Lewis**[1°], **Nicole M. Rankin**[1°]

1 Faculty of Medicine and Health, University of Sydney, Sydney, New South Wales, Australia, 2 Discipline of Genetic Counselling, Graduate School of Health, University of Technology Sydney, Sydney, New South Wales, Australia, 3 Cancer Research Division, Cancer Council NSW, Sydney, New South Wales, Australia

° These authors contributed equally to this work.
‡ These authors also contributed equally to this work.
* rosie.oshea@sydney.edu.au

## Abstract

### Background

Integration of genetic testing into routine oncology care could improve access to testing. This systematic review investigated interventions and the tailored implementation strategies aimed at increasing access to genetic counselling and testing and identifying hereditary cancer in oncology.

### Methods

The search strategy results were reported using the PRISMA statement and four electronic databases were searched. Eligible studies included routine genetic testing for breast and ovarian cancer or uptake after universal tumour screening for colorectal or endometrial cancer. The titles and abstracts were reviewed and the full text articles screened for eligibility. Data extraction was preformed using a designed template and study appraisal was assessed using an adapted Newcastle Ottawa Scale. Extracted data were mapped to Proctor's et al outcomes and the Consolidated Framework for Implementation Research and qualitatively synthesised.

### Results

Twenty-seven studies, published up to May 2020, met the inclusion criteria. Twenty-five studies ranged from poor (72%), fair to good (28%) quality. Most interventions identified were complex (multiple components) such as; patient or health professional education, interdisciplinary practice and a documentation or system change. Forty-eight percent of studies with complex interventions demonstrated on average a 35% increase in access to genetic counselling and a 15% increase in testing completion.

**Data Availability Statement:** All relevant data are within the manuscript and its Supporting information files.

**Funding:** Research funding by a Cancer Council New South Wales PhD scholarship and a

Translational Cancer Research Network Clinical PhD Scholarship Top-up award, supported by the Cancer Institute NSW supports ROS in the completion of her PhD studies in the Faculty of Medicine and Health at The University of Sydney. The funding bodies did not play a direct role in the design of the study or collection, analysis, and interpretation of data or in writing the manuscript.

**Competing interests:** The authors declare that they have no competing interests.

Mapping of study outcomes showed that 70% and 32% of the studies aligned with either the service and client or the implementation level outcome and 96% to the process or inner setting domains of the Consolidated Framework for Implementation Research.

## Conclusion

Existing evidence suggests that complex interventions have a potentially positive effect towards genetic counselling and testing completion rates in oncology services. Studies of sound methodological quality that explore a greater breadth of pre and post implementation outcomes and informed by theory are needed. Such research could inform future service delivery models for the integration of genetics into oncology services.

## Introduction

A challenge of optimising standards in oncology is the slow rate that evidence is adopted into clinical care, leading to inequity and variation between hospital settings [1, 2]. Health services research identifies ways to ease the burden on cancer care provision, improve system inefficiencies and optimise standards [1, 2]. In the case of cancer germline genetic testing (GT), a systematic way to sustain implementation of GT is needed as this is increasingly being used in the assessment and care of patients in many specialities [3]. Evidence based clinical practice guidelines in the United States of America (USA), Australia and the United Kingdom (UK) recommend access for epithelial ovarian cancer (EOC) and triple negative breast cancer (TNBC) patients to have *BRCA* testing [4–6]. Established clinical guidelines for directing access to GT for endometrial and colorectal cancers (EC/CRC) exist in the USA, UK and Australia [7–9].

Direct access to GT in oncology care (known as 'mainstreaming') could improve access to GT and the identification of patients with hereditary cancer. Prior to mainstreaming, access to genetic counselling (GC) services has been through referral to genetics services. In many jurisdictions, medical specialists in oncology can now order a panel of multiple genes to assess for hereditary breast and ovarian cancer (HBOC) [10] without prior referral to genetics services. Mainstreaming assumes that oncology health professionals will take on the role of pre-test GC for GT.

Barriers to mainstreaming exist among non-genetics health professionals from a range of specialities and include, a lack of genetics knowledge and skill, resources and guidelines, low confidence with genetics, and concerns about discrimination and psychological harm [11, 12]. These barriers have led to suboptimal referral and identification of hereditary cancer [13, 14] and reduce the potential for GT to inform cancer prevention through regular screening or preventative surgery [15–17]. Integrating GT into oncology services aims to circumvent recognised barriers to improve the identification of *BRCA* related HBOC and personalise treatments with the use of poly ADP ribose polymerase (PARP) inhibitors (PARPi) [18]. The initial *BRCA* mainstreaming programs allows implementation insights to inform approaches to improve access to GT and identification for other hereditary cancer.

Hereditary colorectal and endometrial cancer associated with Lynch Syndrome (LS) is a parallel example where direct access to GT instead of referral to genetics services allows surgeons and oncologists to directly order GT. Recent changes in Australian public funding of GT [19] in 2020, now allow medical specialists caring for EC and CRC patients to request GT directly, as a new form of mainstreaming. Before ordering GT for CRC or EC patients,

oncologists or surgeons need to identify deficient mismatch repair (dMMR) positive status on a universal tumour screen (UTS) [20]. The aim of UTS is to increase the number of LS individuals identified, enabling cancer screening and risk prevention and reducing the burden of disease in individuals and their families [20].

In Australia, as direct access to GT to align with UTS begins, learning from other jurisdictions where GT has been part of routine oncology care, can provide important lessons. The application of implementation science using Proctor's evaluative framework [21] and the Consolidated Framework for Implementation Research (CFIR) [22] provides a means of assessing existing interventions used to incorporate GT into routine oncology, to understand the effectiveness of mainstreaming strategies and to inform its long-term sustainability.

Understanding implementation outcomes can enhance the implementation success of an intervention. However, many studies miss out this important step, focusing the evaluation of the interventions' implementation on the service and client level [21]. Therefore, an implementation outcome evaluative framework provides a means to assess and evaluate implementation efforts, differentiating three groups of outcomes–implementation, service and client [21]. The CFIR framework [22] allows an understanding of the factors that can affect implementation processes and outcomes.

Interventions are most effective when there is an understanding of the constituent components, implementation factors in the relevant health system and the implementation outcomes of the intervention [21, 23]. For the purposes of this review an intervention is defined as a single unit that can bring about change in a system [23] and complex interventions are described as 'interventions that contain several interacting components' [24]. The term complex refers to the multi component nature of the health system intervention and relates to the intervention, setting, patients and professionals interacting with it [24]. An example of a single unit intervention would be education about incorporating GT into routine oncology practice. A complex intervention example would consist of multiple components, for example, education/training of staff, changes to referral pathways and use of electronic medical record to streamline appointments. These components, which can be described as 'implementation strategies', strive to increase access to GC and GT in routine oncology practice. The specific review question we asked was: What interventions have been shown to increase the uptake of GC and GT in oncology services, specifically for ovarian, breast, colorectal and endometrial cancer, to identify hereditary cancer? Interventions of interest were those that aimed to:

1. increase GT integration in oncology care (mainstreaming) for subsets of ovarian and breast cancer in the oncology setting, and

2. increase the uptake of GT after UTS for colorectal and endometrial cancer.

    Our outcomes of interest for intervention studies were:

    1. Referral rates of eligible patients with breast, ovarian, endometrial and colorectal cancer to GC

    2. Breast, ovarian, colorectal and endometrial cancer patients completing GC and GT

    3. Identification of hereditary cancer.

    The second objective was to understand the implementation factors that influence GT adoption in oncology services.
    Our outcomes of interest for implementation factors were;

    4. Qualitative or quantitative implementation outcome factors.

## Methods

This systematic review used the Preferred Reporting Items for Systematic Reviews and Meta-Analyses statement (PRISMA) [25] to report the search results. The protocol of the review was not registered as it is part of a PhD program of study.

### Inclusion/exclusion criteria

Study inclusion criteria were as follows;

1.  A population consisting of;

    - Breast, ovarian, colorectal and endometrial cancer patients > 18 years old with 80% of the population being studied for access to GT for HBOC or uptake of GT after UTS for CRC or EC

2.  An intervention focussed on the following;

    - integration of routine genetic testing through mainstreaming for breast and ovarian cancer in oncology services

    - increasing GC and GT completion rates after UTS for CRC and EC

3.  A comparator consisting of the following;

    - Another intervention with the same purpose described in intervention section above

    - No intervention (in the case of qualitative studies)

    - Standard or usual care

4.  Outcomes focusing on Proctor's evaluative framework and CFIR's five implementation factor domains as follows;

    - Implementation outcomes and factors

    - Service outcomes and factors

    - Client outcomes and factors

5.  Study designs as specified below;

    - Experimental, quasi-experimental and observational study designs (randomised control trials, cohort studies, controlled pre and post studies, case series).

    - Qualitative studies on implementation factors or outcomes that influence genetic testing adoption in oncology

6.  Organisation setting;

    - Any healthcare system engaging in integrating GT into oncology services.

A study was excluded if it focused on patients with other cancers not related to HBOC and LS or asymptomatic individuals or relatives at high risk of these conditions. Additionally, a study was excluded if the outcomes were not linked to mainstreaming of GT or enhancing the uptake of UTS.

## Search strategy

A search strategy was developed by checking the subject headings and text terms used for the area of interest. An initial draft was reviewed with systematic review experts (CC SH) and trialled on MEDLINE. The search terms were revised by ROS and systematic reviewers (CC) and the final version included search terms for ovarian, breast, colorectal and endometrial cancer, combined with genetic counselling, genetic testing, mainstreaming, and implementation science terms (S1 Table). This strategy was then translated for use in CINAHL (S2 Table). The strategy was executed in four databases on 26.09.19: MEDLINE, EMBASE, PsychINFO and CINAHL and alerts from this search were screened until 26.05.20 A list of included studies is in the Supporting information.

## Study selection

All of the titles and abstracts were exported to Endnote X8 and screened by ROS independently against the inclusion criteria. Full text articles of those with unclear or missing information were retrieved and screened by ROS against the inclusion criteria. Studies meeting the inclusion criteria were retained. ROS obtained all relevant full texts articles and randomly assigned these to two other reviews through Excel. Full text articles were screened for eligibility by three reviewers (ROS, AC and CC) and the reasons for excluding articles were documented in Excel. Any disagreements were resolved through initial discussion between the three reviewers and if no consensus was reached, a fourth reviewer was considered an arbitrator.

## Data extraction

Data was extracted from each included study on population (healthcare professional, setting and patient); description of the intervention (adapted criteria template for intervention description and replication (TIDieR) checklist [26]); implementation study dates, use of a model or framework, study design and intervention outcomes for the implementation period were assessed by mapping to all of the components of Proctor et al.'s evaluative framework [21] at the implementation, service or client level and all CFIR [22] domain and relevant constructs; and information for quality appraisal (S4 Table). Two reviewers (ROS, CC) extracted data from two included studies and compared results. Discrepancies were discussed and a consensus reached for future data extraction. One reviewer (ROS) independently completed data extraction for the remainder of the included studies (S4 Table).

## Quality appraisal

The quality assessment of each included study was assessed using an adapted Newcastle Ottawa Scale (NOS) [27] for cohort studies and further adapted for case series with implementation outcomes. The adapted NOS evaluated selection bias, study design, confounders, blinding of study participants, data collection methods, and follow up rates. The tool was independently applied by two reviewers (AC, ROS) and consensus was reached on any discrepancies through discussion. The star rating for each component was then converted to Agency for Healthcare Research and Quality (AHRQ) rating from poor to good quality. Qualitative studies were assessed using the Critical Appraisal Skills Programme (CASP) [28].

## Data synthesis

Descriptive data analysis was preformed to summarise study characteristics through proportions or percentages on study design and location, types of health professionals and patients targeted with the intervention, the hospital setting used and quality appraisal. Heterogeneity in

**Table 1. Classification of intervention components into four distinct domains.**

| Complex intervention [24] type | Implementation strategies [23] |
|---|---|
| **Education (health professional or patient)** | Face to face education |
| | Online education |
| | Written information |
| | Family history collection proforma |
| **Interdisciplinary practice** | Genetic counsellor at multidisciplinary team (MDT) meeting |
| | Embedded Genetic counsellor in oncology |
| | Genetic counsellor or oncologist facilitates communication |
| | Genetics or oncologist led referral pathway |
| | Patient navigators |
| **Documentation (GC referral, GT outcomes and written information to facilitate mainstreaming)** | Use of electronic medical record (EMR) or MDT proforma |
| | Testing protocol |
| | Pathway or checklist |
| | Standardised letters for results |
| | Consent form |
| **Systems (electronic or process)** | Smart text for EMR or pathology reporting |
| | Synchronous scheduling of GC appointments |
| | Shared GC referral or review e-mail inbox |
| | E-mail alerts |
| | E-mail notifications for referral |
| | EMR GC referral |
| | Result tracking |

*MDT* multidisciplinary team, *EMR* electronic medical record, *GC* Genetic Counselling, *GT* Genetic testing.

intervention characteristics, measured outcomes, and small sample sizes did not allow for a meta-analysis. A narrative synthesis was performed to summarise and explain the intervention characteristics and potential effects. The intervention effectiveness (absolute difference) was measured for studies with intervention and control data (S4 Table). Due to the heterogeneity of intervention components, a domain directed intervention classification system was created (Table 1). We used the designed intervention classification system (Table 1), Proctor's evaluative framework [21] and CFIR [22] as the sensitising lens for thematic analysis. Each study's outcomes were mapped to Proctor's implementation, service and client outcomes and implementation factors through CFIR's domains and constructs. The development of themes and subthemes was informed from this overarching structure. Each study was checked and referred to as per the disease context (breast and ovarian cancer versus colorectal and endometrial cancer) as themes were incorporated into a narrative synthesis. Three reviewers (AC, NR, CJ) commented on and discussed a draft of the themes and sub-themes, and a final version was agreed.

## Results

### Study characteristics

**Studies and location.** Of the 2224 titles generated through database-searching (Fig 1), we included 27 [29–55] studies of which 25 [29–48, 51–55] described interventions. The majority

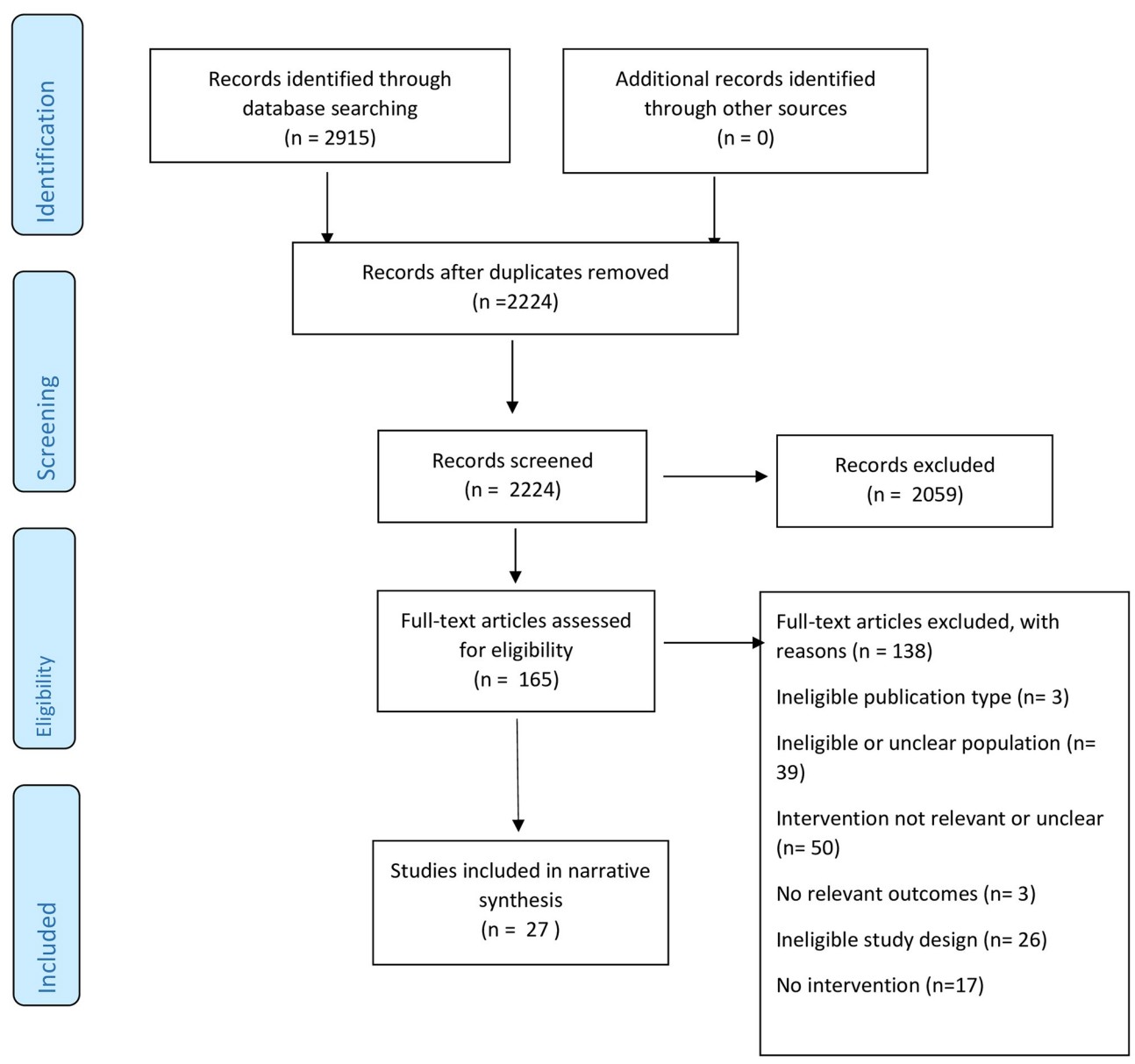

**Fig 1. Flow chart summarizing identification of studies for inclusion in this systematic review using PRISMA [25].**

of included studies (44%) were from North America [34–39, 41, 44–47, 54] (Table 2). The 25 studies [29–48, 51–55] (93%) described interventions to increase access to GC and GT through mainstreaming or UTS initiatives (S4 Table). The study designs found were retrospective or prospective cohort studies with concurrent or historical controls (44%) [34, 36, 37, 42–47, 54, 55] or case series that reported on intervention outcomes (56%) [29–33, 35, 38–41, 48, 51–53] (Table 2). Two qualitative studies [49, 50] and eight of the intervention studies (with a qualitative or quantitative component) [29–31, 33, 40, 48, 51, 54] described implementation outcomes that relate to acceptability and cost of interventions (Table 5, S4 Table).

**Participants.** Twenty-one studies included a variety of healthcare professionals (years of practice not indicated) exposed to the interventions (Table 2) and four studies did not specify the health professionals involved. The numbers of patients exposed to the intervention in the

**Table 2. Summary of included studies and participants' characteristic.**

| Health professionals targeted | Breast and ovarian cancer intervention (n = 20) | | Colorectal and endometrial cancer intervention (n = 5) | |
|---|---|---|---|---|
| | Number of studies/Total (%) | References | Number of studies/Total (%) | References |
| Genetic counsellors | 12/20 (60%) | [29, 31, 33, 35, 37, 38, 41, 43, 46, 48, 53, 54] | 3/5 (60%) | [42, 44, 47] |
| Medical oncologists | 13/20 (65%) | [29, 30–33, 37, 43, 45, 48, 51, 53–55] | 2/5 (40%) | [42, 47] |
| Gynaecology oncologists | 9/20 (45%) | [29, 33, 35, 38, 39, 45, 46, 53, 55] | NA | |
| Clinical nurse specialists | 5/20 (25%) | [29, 30, 33, 37, 55] | 3/5 (60%) | [39, 41, 42] |
| Advanced nurse practitioners | 3/20 (15%) | [35, 38, 39] | NA | |
| Clinical geneticists | 4/20 (20%) | [37, 43, 48] | 2/5 (40%) | [42, 47] |
| Resident or fellow or trainee | 2/20 (10%) | [35, 37] | | |
| Other | 1/20 (5%) | [38, 45] | | |
| Pathologists | NA | | 5/5 (100%) | [39, 41, 42, 44, 47] |
| Surgeons | NA | | 4/5 (80%) | [41, 42, 44, 47] |
| **Location all studies n = 27** | | | | |
| North American and Canada | 12/27 (44%) | [34–39, 41, 44–47, 54] | | |
| United Kingdom | 8/27 (30%) | [29, 30, 31, 32, 40, 50, 51, 52] | | |
| Australia | 5/27 (18%) | [33, 48, 43, 42, 49] | | |
| Europe | 2/27 (7%) | [53, 55] | | |
| **System setting n = 25** | | | | |
| Single site urban hospitals | 16/25 (59%) | [29, 30, 33, 36, 37, 39, 40, 43, 46, 47, 51, 52, 55] | | |
| Large multi-site urban and regional hospital | 6/25 (24%) | [34, 35, 38, 41, 44, 53] | | |
| State wide systems | 3/25 (12%) | [43, 45, 54] | | |
| Unspecified health system | 1/25 (4%) | [42] | | |
| **Study Design n = 27** | | | | |
| Cohort | 11/27 (40%) | [34, 36, 37, 42–44, 46, 47, 54, 55] | | |
| Case series | 14/27 (51%) | [29–33, 35, 38, 39–41, 48, 51–53] | | |
| Qualitative | 2/27 (7%) | [49, 50] | | |

studies ranged from 16 to 1214. Nearly half of the studies (44%) [30, 32, 33, 36, 37, 41, 43, 46, 47, 51, 54] had fewer than 200 patients exposed to the intervention. Seventeen studies (68%) reported participants' ages [29, 31–33, 35–37, 39–41, 45, 46, 51–55] and 14 (56%) reported subtypes of cancer [29, 31–33, 35, 37–40, 43, 45, 46, 51, 52].

**Interventions, setting and framework.** The majority of studies used complex interventions (Table 1, S4 Table) to increase access to GC and GT, either in the routine oncology setting [29–40, 51–55] or optimizing referral to genetic services for GC and GT [45–47] for ovarian or breast cancers and through optimizing access to genetic services after UTS in colorectal and endometrial cancer [39, 41–44]. The 25 studies spanned a variety of health systems (Table 2) with six studies (24%) included either a quality improvement or process model [35–37, 39, 40] or an implementation science framework [42] to guide implementation. None of the studies used an evaluation framework to underpin the outcomes with a robust assessment of intervention effectiveness.

**Quality assessment.** Fifty-six percent of the studies (n = 14) received a poor AHRQ rating due to the study design–case series with no comparator [29–33, 35, 38–41, 48, 51–53], selection bias in the use of a single site health system [29, 30, 39, 40, 47, 48] and/or no statistical

adjustment for patient population differences or assessment of confounders [30, 42, 47] (S5 and S6 Tables).

Thirty-six percent of the studies received a fair to good AHRQ rating (n = 9) and were cohort studies with a historical or concurrent comparator [34, 36, 37, 44–47, 54, 55]. Statistical analysis was preformed between intervention and control but were not adjusted for differences in patient population characteristics or confounders, apart from two studies [46, 54] that performed regression analysis (S4 Table). All studies except two [44, 45] had >80% of the patient population followed up in the study period.

Ninety-one percent (10/11) of historical or concurrent cohort studies had between 10 months to three and a half years when the intervention was implemented [34, 36, 41–46, 54, 55], allowing sufficient follow up time for outcomes to be measured. The two qualitative studies [49, 50]. assessed using the CASP tool [28] reached a high-quality rating score with all questions (1–10) addressed in each study.

**Mapping of outcomes and studies to framework.** About two-thirds of studies (64%) measured the following outcomes to assess the adoption of the intervention at the service level; GC recommendation and referral, GC and GT completion rate and at the client level, through identification of hereditary cancer (68%) and treatment management impact (Table 3). About one-third of studies measured implementation level outcomes, acceptability through satisfaction with the intervention (32%) and cost effectiveness (16%). Studies focussed on the process domain of CFIR in relation to engaging with health professionals in the implementation effort (96%) and on executing (24%) using a process model or implementation framework to execute the implementation plan (Table 3). The available resource construct of the inner setting domain mapped to 96% of studies using the health professional as the resource for implementation efforts and access to knowledge and information about the intervention (64%), through education as a core component of the intervention. Twelve percent of studies mapped to the process domain-reflecting and evaluating through health professional's feedback about the intervention. Characteristics of individuals—self-efficacy (16%) and outer setting—patients' needs and resources (n = 6) or intervention characteristics–cost (16%) were also addressed (Table 3).

## Intervention outcomes and implementation factors

The following themes describe the potential effects of complex interventions for the integration of GT in ovarian, breast, colorectal and endometrial cancer settings along with outcomes and factors at the implementation, service and client level.

### Increasing access to genetic counselling and genetic testing in routine oncology for ovarian and breast cancer

Twenty-five studies described interventions to increase access to GC and completion of GT in breast and ovarian cancer patients through; referral rates to GC [29, 34–39, 46–48, 52, 53, 55], GC [34–40, 46–48, 51–54] or GT completion [31, 34–40, 47–48, 51–55], identification of hereditary cancer [29, 43, 31–36, 40, 48, 51, 52, 54, 55], time to gain access to GT and results [29, 31–33, 35, 45, 52, 54], treatment management impact [29, 33, 52, 55] and uptake of predictive testing in families [29, 32, 45] (S4 Table). The implementation strategies used were varied and classified under complex intervention type of education, documentation, interdisciplinary practice or electronic systems domains (Table 1). Twenty studies mapped to Proctor's evaluative framework [21] at the service or client level measuring effectiveness through the outcomes outlined above [29–40], [43–46, 51–55] (Table 3). CFIR [22] mapped to implementation process factors through the executing and engaging constructs with five studies using a quality

**Table 3. Proctor et al.'s implementation outcome framework [21] and CFIR [22] applied to outcomes of included studies.**

| Domain | Description | Measure | No. of intervention studies/Total (%) |
|---|---|---|---|
| **Implementation outcomes and factors** | | | |
| *Proctor et al.* [21]. *(hereafter 'Proctor')* Implementation outcomes • the effects of deliberate and purposive actions to implement new treatments, practices, and services [21] | Acceptability The perception among implementation stakeholders that a given treatment, service, practice, or innovation is agreeable, palatable, or satisfactory [21] | • Patients or healthcare professionals' satisfaction with the mainstreaming intervention [29–31, 33, 40, 48, 51, 54] | 8/25 (32%) |
| | Cost The cost impact of an implementation effort [21] | • implementation cost of intervention or cost savings [29, 31, 40, 48] | 4/25 (16%) |
| *CFIR* Process | Reflecting & Evaluating Quantitative and qualitative feedback about the progress and quality of implementation accompanied with regular personal and team debriefing about progress and experience [22] | • Healthcare professionals' feedback about the intervention [40, 42, 54] | 3/25 (12%) |
| Self-efficacy | Characteristics of individuals Individual belief in their own capabilities to execute courses of action to achieve implementation goals [22] | • Healthcare professionals' belief about their ability to undertake intervention [29, 30, 33, 40] | 4/25 (16%) |
| Intervention Characteristics | Cost Costs of the innovation and costs associated with implementing the innovation including investment, supply, and opportunity costs [22] | • implementation cost of intervention or cost savings [29, 31, 40, 48] | 4/25 (16%) |
| **Service outcomes and implementation factors** | | | |
| *Proctor* Service Outcomes • the extent to which services are safe, effective, patient -centred, timely, efficient, and equitable [56, 57] | Effectiveness Providing services based on scientific knowledge to all who could benefit [56, 57] | • GC Referral [29, 34–39, 41, 42, 44, 46–48, 52, 53, 55] • GC completed [34–41, 44, 46–48, 51–54] • GT completed [31, 34–41, 44, 47–48, 51–55] • Patients with identified gene mutations [29, 43, 31–36, 39, 40, 41, 44, 48, 51, 52, 54, 55] | 16/25 (64%) |
| | Timeliness Reducing waits and sometimes harmful delays for both those who receive and those who give care [56, 57] | • Time to GC or GT [32–35, 52] and results [29, 31–33, 54] | 10/25 (40%) |
| | Equity Providing care that does not vary in quality because of personal characteristics [56, 57] | • GT access and undertaken [31, 34–41, 44, 47–48, 51–55] • GC referrals [29, 34–39, 41, 42, 44, 46–48, 52, 53, 55] | 17/25 (68%) 16/25 (64%) |
| *CFIR* Process | Executing Carrying out or accomplishing the implementation according to plan [22] | • use of a quality improvement or process model [35–37, 39, 40] • use of an implementation science framework [42] | 6/25 (24%) |
| | Engaging Attracting and involving appropriate individuals in the implementation and use of the intervention through a combined strategy of social marketing, education, role modelling, training, and other similar activities [22] | • Engaging health professionals through education or implementing the intervention [29, 30–48, 51, 52, 54, 55] | 24/25 (96%) |
| Inner setting | Readiness for Implementation–Available resources The level of resources dedicated for implementation and on-going operations, including money, training, education, physical space, and time [22] | • use of health professional as a resource for implementation [29–48, 51, 52, 54, 55] | 24/25 (96%) |
| | Access to Knowledge & Information Ease of access to digestible information and knowledge about the intervention and how to incorporate it into work tasks [22] | • use of education as a component of the intervention [29, 30, 32, 33, 36–40, 42, 45, 46, 51–54] | 16/25 (64%) |

*(Continued)*

**Table 3.** (Continued)

| Domain | Description | Measure | No. of intervention studies/Total (%) |
|---|---|---|---|
| **Client outcomes and implementation factors** | | | |
| *Proctor*<br>Client Outcomes<br>Consumer wellbeing and clinical effectiveness [21] | Satisfaction<br>The consumers' satisfaction with the intervention used [21] | • Patients satisfaction with mainstreaming intervention [29–31, 48, 51, 52] | 6/25 (24%) |
| | Symptomology<br>Identifying hereditary cancer so that patients and health professionals can enact treatment management and cancer prevention strategies | • Identification of hereditary cancer [29, 43, 31–36, 39–41, 44, 48, 51, 52, 54, 55]<br>• Access or referral to cancer prevention information [29, 32, 45, 55]<br>• Treatment management impact [29, 33, 52, 55] | 17/25 (68%)<br>4/25 (16%)<br>4/25 (16%) |
| *CFIR*<br>Outer setting | Patient Needs & Resources<br>The extent to which patient needs, as well as barriers and facilitators to meet those needs, are accurately known and prioritized by the organization [22] | • Patients satisfaction with mainstreaming intervention [29, 30, 31, 48, 51, 52] | 6/25 (24%) |

*GC* Genetic Counselling *GT* Genetic testing.

improvement or process model [35–37, 39, 40] carrying out implementation according to a plan and 20 studies engaging health professionals through education or implementing the intervention [29–40, 43, 45, 46, 48, 51, 52, 54, 55] (Table 3).

Nine studies showed a potential positive effect in favour of the intervention having an impact mainly on GC referral [34, 36, 38, 43, 44, 46, 55] and GC and/or GT completion [34, 36, 38, 43, 45, 46, 55]. Thirteen studies revealed an unclear intervention impact in relation to all outcomes with no comparator presented for assessment [29, 31, 32, 33, 35, 39, 40, 47, 48, 51–54].

**Complex interventions—Education, documentation and electronic system.** Four studies [36, 37, 45, 46] used complex interventions related to education, documentation and electronic system domains. One of these studies [36], employed a complex intervention consisting of education (patient and clinicians), documentation (smart text and written handouts in EMR and for patient) and electronic system (EMR documentation of GC referral and outcome in MDT, GC appointment scheduling). The outcomes assessed showed a significant difference in GC referral rates (+51.2 (95% CI 43.9–58.5) p ≤0.001), patients completing GC (+54 (95% CI 45.3–62.8) p ≤0.001) and GT (+13.2 (95%CI 3.3–23.3) p = 0.007), between the intervention and control [36]. Another study [37] used a complex intervention consisting of health professional and patient education (written information, family history collection proforma), documentation (EMR documentation of referral for GC/GT and testing protocol pathway) and systems (scheduling GC appointments directly at gynaecology clinic) showed a trend towards the intervention for GC referral (+27.4 (95% CI 11.1–43.7) p = 0.02) and completion of GT (+20.6 (95% CI 5.9–35.4) and towards the control for completion of GC (-27.8 (95% CI -46.7 to -9.1)) and identifying hereditary cancer (- 17.9 (95% CI– 40.9–5.1) p = 0.17). This study was limited by a small study sample size and short follow up period.

Two studies [45, 46] employed an education, documentation and systems complex intervention. The education (clinicians educated on EOC GC referral guidelines) and systems (use of smart text to refer all EOC to GC on the pathology report) intervention showed an absolute difference in eligible serous histology patients completing GC and GT (+13.7% (95% CI 7.6–19.1) [45] (Table 4).

**Table 4. Implementation strategies in the various health system and professional settings grouped by complex intervention effects results.**

| Reference | Population and setting | Implementation strategies | Absolute Difference % | Framework mapping | Study Quality and design |
|---|---|---|---|---|---|
| **Group 1: Results significantly favour complex intervention** | | | | | |
| Uyar [36] 2018 USA | *Healthcare Professionals*: All gynaecology oncology providers non-specified *Patients*: All women with EOC *Healthcare Institution*: Academic cancer centre | Education for patients and healthcare professionals Handouts for patients EMR documentation of GC/GT and/or referral GC at MDT or documentation of GC and GT outcomes Scheduling GC appointments directly at gynaecology clinic Rates of GC/GT recommendation in EHR (Electronic Health Record) | Outcome 1. Rates of GC/GT recommendation in EHR + 67.7% (95% CI 59.8–75.6) p value not provided Outcome 2. GC referral +51.2% (95% CI 43.9–58.5, p ≤0.001) Outcome 3. GC completion +54% (95% CI 45.3–62.8, p ≤0.001) GT completion +13.2% (95% CI 3.3–23.3, p = 0.007) Outcome 4 Patients identified with *BRCA* mutations + 3.6% (95% CI -9.4–16.5, p = 0.68) | *Service*: Effectiveness • GC referral • GC completion • GT completion • Patients with identified gene mutations Equity • GT access • GC referrals • GT undertaken *Client*: Cancer prevention • Identification of hereditary Cancer **CFIR** *Inner setting* Readiness for implementation • access to knowledge and information *Process* Engaging • key stakeholders Executing | Fair Quality Cohort study with historical control Single site health system and no analysis on confounding variables or regression analysis on the characteristics inherent in the control verses the intervention population or health system |
| Brown [38] 2018 USA | *Healthcare Professionals* Gynaecology oncologists Breast surgeons Genetic counsellors Patient navigators Advanced care providers *Patients*: All women with EOC Triple Negative Breast Cancer < 60years Breast Cancer < 45 years *Healthcare Institution*: Comprehensive not-for-profit system with more than 900 care locations in 2 states, including academic medical centres, hospitals, freestanding emergency departments, health care pavilions, physician practices, and outpatient surgical centres. | Patient navigators in gynaecologic oncology and breast surgery clinics. Increase volume of GC and telemedicine consults Education to all gynaecologic oncologists, breast surgeons, and advanced care providers on guidelines Referral to GC was made a standard of practice | Outcome 1. GC referral EOC +59.7% (95% CI 50.2–69.4, p<0.05) TNBC < 60 yrs +21.2% (95% CI 10.6–31.8, p<0.05) BrCa < 45 yrs +6.3% (95% CI -1.0–13.5) p value not provided Outcome 2. GT completion EOC +29% (95% CI 16.8–41.2, p<0.05) TNBC < 60 yrs +26.6% (95% CI 14.9–38.4, p<0.05) BrCa < 45 yrs +15.7% (95% CI -7.5–6.1, p<0.05) Outcome 3. Patients identified with *BRCA* mutations EOC +7.5% (95% CI– 7.9–23, p = 0.53) TNBC < 60 yrs +0.22% (95% CI -8.2–12.6) p value not provided BrCa < 45 yrs -0.54% (95% CI -7.2–6.1) p value not provided | *Service*: Effectiveness • GT undertaken • GC referrals Equity • GT access • GC referrals • GT undertaken *Client*: Cancer prevention • Identification of hereditary Cancer **CFIR** *Inner setting* Readiness for implementation • access to knowledge & information • available resources *Process* Engaging • key stakeholders | Poor Quality Case series with no comparator to control |

*(Continued)*

**Table 4.** (*Continued*)

| Reference | Population and setting | Implementation strategies | Absolute Difference % | Framework mapping | Study Quality and design |
|---|---|---|---|---|---|
| Miesfeldt [41] 2018 USA | *Healthcare Professionals*: Pathologist Surgeon Patient navigator—Oncology Nurse *Patients*: All colorectal and uterine cancer *Healthcare Institution*: Medical Centre Cancer Institute's Cancer Risk and Prevention Clinic— community hospital and a state tertiary centre with a GC-supported cancer genetic program | Triggered GC referral after abnormal IHC and MSI Pathology communication via e-mail to surgeon Patient navigator to ensure follow through to GC for abnormal IHC and MSI | Outcome 1. GC referral I: 16/16 (100.0) C:12/12 (100.0) p value not provided Outcome 2. GC completion +45.8% (95% CI 13.6–78.1, p = 0.020) Outcome 3. GT completion +12.9% (95% CI -24.7–50.4) p value not provided Outcome 4. Patients identified with *BRCA* mutations +28.8% (95% CI -21.5–79.2) p value not provided | *Service*: Effectiveness • GT undertaken • GC referral • GC apt uptake *Equity* • GT access • GC referrals • GT undertaken *Client*: Cancer prevention • Identification of hereditary Cancer **CFIR** *Inner setting* Readiness for implementation • available resources *Process* Engaging • key stakeholders | Poor quality Case series with no comparator for control |
| Heald [44] 2013 USA | *Healthcare Professionals*: Genetic Counsellor Colorectal Surgeon Pathologist *Patients*: All patients with colorectal cancer *Healthcare Institution*: Academic and tertiary (2 regional community hospitals) and primary care centres (multiple family health centres) | Triggered EMR GC referral after abnormal IHC and MSI to surgeon EMR documentation of GC/GT and/or referral via email GC embedded to increase communication of abnormal IHC to patients and facilitate referral Shared GC e-mail to review all abnormal MSI and IHC from pathologist to GC | Outcome 1. GC referral GC v No GC +44.7% (95% CI 28.1–60.5, p<0.001) GC & Surgeon v No GC +26.5% (95% CI -1.2–54.2, p = 0.023) Outcome 2. GC completion GC v No GC +39.8% (95% CI 20.9–58.8, p<0.001) GC & Surgeon v No GC +32.0% (95% CI 0.017–64) p value not provided Outcome 3. GT completion GC v No GC +39.8% (95% CI 21.1–58.5, p<0.001) GC & Surgeon v No GC +19.2% (95% CI -13.4–51.7) p value not provided Outcome 4. Patients identified with LS GC v No GC +22.5% (95% CI 7.7–37.2) GC & Surgeon v No GC +1.2% (95% CI -17.8–20.2) p value not provided Outcome 5. Time to appointment GC v No GC1–413 days p<0.001 GC & Surgeon v No GC -164 days p value not provided | *Service*: Effectiveness • GT undertaken • GC referral • GC apt uptake Timeliness • Time to GC apt *Equity* • GT access • GC referral • GT undertaken *Client*: Cancer prevention • Identification of hereditary Cancer **CFIR** *Inner setting* Readiness for implementation • available resources *Process* Engaging • key stakeholders | Fair Quality Cohort study with historical control Single site health system with no analysis on confounding variables or regression analysis on the characteristics inherent in the control verses the intervention population or health system Less than 80% of population followed up |
| Senter [34] 2017 USA | *Healthcare Professionals*: Gynaecology oncology and cancer genetics health professionals- unspecified *Patients*: All women with EOC *Healthcare Institution*: Large academic medical comprehensive cancer centre | GC embed in oncology services EMR documentation of GC/GT and/or referral Scheduling GC appointments directly at gynaecology clinic | Outcome 1. GC referral +22.8% (95% CI 16.7–29.4, p<0.00001) Outcome 2. GC completion +45.5% (95% CI 33.6–57.6, p<0.00001) Outcome 3. Time to gain access to GC I: 1.67 months C:2.52 months P< 0.01 | *Service*: Effectiveness • GC referral • GC and GT completion *Equity* • GT access • GC referrals • GT undertaken Timeliness • Time to GC apt **CFIR** *Inner setting* Readiness for implementation • available resources *Process* Engaging • key stakeholders | Good quality Cohort study with historical control |
| **Group 2: Results trend towards complex or single unit intervention** | | | | | |

(*Continued*)

**Table 4.** (Continued)

| Reference | Population and setting | Implementation strategies | Absolute Difference % | Framework mapping | Study Quality and design |
|---|---|---|---|---|---|
| Hanley [45] 2018 USA | *Healthcare Professionals*: Family practitioners General obstetrician Gynaecologists Medical and gynaecology oncologists *Patients*: All patients with serous, endometroid and clear cell ovarian cancer type *Healthcare Institution*: State wide Hereditary cancer program | Education to healthcare professionals on GC and GT referral guidelines for ovarian cancer Smart text including standard recommendation to refer to GC included on the pathology report | Outcome 1. GC and GT completion by histopathology Serous +13.7% (95% CI 7.6–19.1) (OR = 4.70; 95% CI 2.89–7.62) Endometrioid -6.3% (95% CI -6.4 to– 2.4) Clear cell -3.3% (95% CI -6.2 to -0.4) Unknown -4.2% P< 0.001 serous vs endometroid and clear cell cancers getting GT after 2010 Outcome 2. Patients identified with *BRCA* Serous histopathology +6.2% (95% CI -6.1 to 19.4, P = 0.519) Outcome 3. Cancer prevention Familial predictive GT uptake and mutation identification Carrier tests +0.73% p = 0.071 Family members identified as *BRCA* +0.56% p = 0.009 Carrier tests per serous histopathology + 0.76% P = 0.098 Family members identified as *BRCA* positive +0.65% P = 0.012 | *Service*: Effectiveness • GT undertaken • GC uptake Equity • GT access • GT undertaken *Client*: Cancer prevention • Identification of hereditary Cancer **CFIR** *Inner setting* Readiness for implementation • access to knowledge and information *Process* Engaging • key stakeholders | Fair to poor quality Cohort study with historical control Multisite health system but with no analysis on confounding variables or regression analysis on the characteristics inherent in the control verses the intervention population or health system Unclear how many patients were followed up |
| Petzel [46] 2014 USA | *Healthcare Professionals*: Gynaecology oncologists Genetic Counsellor *Patients*: All women with EOC *Healthcare Institution*: Primary academic metro Women's Cancer Centre | EMR referral to GC EMR documentation of GC referral Use of referral guidelines and checklist | Outcome 1. GC referral +12.7% (95% CI -0.04–25.4, P = 0.053) Outcome 2. GC completion +9.9% (95% CI– 0.41–20.4) p value not provided | *Service*: Effectiveness • GC referrals • GC uptake Equity • GT access • GC referral • GT undertaken **CFIR** *Inner setting* Readiness for implementation • access to knowledge and information *Process* Engaging • key stakeholders | Good quality Cohort study with historical control Single site with regression analysis on the characteristics inherent in the control verses the intervention population or health system but no analysis on confounding variables |
| Cohen [43] 2016 Australia | *Healthcare Professionals*: Geneticist Genetic Counsellor Oncologists *Patients*: All patients with EOC < 70 years old *Healthcare Institution*: Metropolitan hospital | Genetics attendance at an MDT tumour board meeting in gynaecology oncology | Outcome 1. GC referral +25% (95% CI 13.6–36.4, P < 0.0001) Outcome 2. GC completion -7.4% (95% CI– 16.8 to 1.9) p value not provided GT completion -16% (95% CI -32.9 to– 0.14) p value not provided Outcome 3. Patients identified with *BRCA* mutations +1.9% (95% CI -22.9–26.9) p value not provided | *Service*: Effectiveness • GC referral • GT undertaken Equity • GT access • GC referral • GT undertaken *Client*: Cancer prevention • Identification of hereditary Cancer **CFIR** *Process* Engaging • key stakeholders | Fair Quality Cohort study with historical control State-wide health system with no analysis on confounding variables or regression analysis on the characteristics inherent in the control verses the intervention population or health system |

*(Continued)*

**Table 4.** (Continued)

| Reference | Population and setting | Implementation strategies | Absolute Difference % | Framework mapping | Study Quality and design |
|---|---|---|---|---|---|
| Lobo [55] 2018 Spain | *Healthcare Professionals*<br>Medical oncologist<br>Cancer Nurse<br>Psychologist<br>General Surgeon<br>Gynaecologist<br>*Patients*:<br>Breast cancer patients<br>*Healthcare Institution*:<br>Single site urban hospital, Madrid Spain | Oncologist led pathway and communication<br>MDT oncology led | Outcome 1. Eligible for GC referral +0.97% (95% CI -3.3–5.3) p value not provided<br>Outcome 2. GC referral +25.4% (95% CI 16.4–34.3, p < 0.0001)<br>Outcome 3.GT completion -11% (95% CI -23.3–0.069) p value not provided<br>Outcome 4. Patients identified with *BRCA* mutations -5% (95% CI -18–8) p value not provided<br>Outcome 5. Cancer prevention management impact +22% (95% CI -16.2–60.3, p = 0.03) | *Service*<br>Effectiveness<br>• GC referral<br>• GC completion<br>• GT completion<br>*Client*<br>Equity<br>• GT access<br>GC referral<br>Cancer prevention<br>• Identification of hereditary Cancer<br>• cancer prevention strategies up taken<br>**CFIR**<br>*Inner setting*<br>Readiness for implementation<br>• available resources<br>*Process*<br>Engaging<br>• key stakeholders | Fair Quality Cohort study with historical control<br>Single site health system and no analysis on confounding variables or regression analysis on the characteristics inherent in the control verses the intervention population or health system<br>Unclear how many patients followed up |
| **Group 3: Results with unclear complex intervention effect** | | | | | |
| George [29] 2016 UK | *Healthcare Professionals*:<br>Gynaecology oncologist<br>Specialist nurse<br>Medical oncologist<br>Genetic Counsellor<br>*Patients*: All women with EOC<br>*Healthcare Institution*:<br>Publicly funded cancer unit at a major treating centre | Education for healthcare professionals<br>Testing protocol pathway<br>Handouts for patients and healthcare professionals<br>Standardised letters for results<br>Standardised consent form | Outcome 1. GC and GT referral I: 207/207 100% C: NR p value not provided<br>2.Time to gain access to genetic test results I: Four-fold reduction in time to result C:NR p value not provided<br>Outcome 3. Patients identified with *BRCA* mutations I: 33/207; 16% C: NR p value not provided<br>Outcome 4. Treatment management I:132/207 (64%) 20/23 *BRCA*+—PARPi access C: NR I: 31/32 with mutations breast cancer surveillance C: NR p value not provided | *Implementation*:<br>Acceptability<br>• Satisfaction with mainstreaming intervention<br>Cost<br>• implementation cost<br>*Service*:<br>Efficiency<br>• Time to gain access to GT<br>Effectiveness<br>• GC referral<br>• Patients with identified gene mutations<br>*Equity*<br>• GT access<br>• GC referral<br>Patient centeredness<br>• Patients satisfaction with mainstreaming intervention<br>*Client*:<br>Cancer prevention<br>• Identification of hereditary Cancer<br>• Access to cancer prevention information<br>• Referral for cancer prevention<br>**CFIR**<br>*Intervention Characteristics*<br>• Cost<br>*Inner setting*<br>Readiness for implementation<br>• access to knowledge and information<br>*Process*<br>Engaging<br>• key stakeholders<br>*Characteristics of Individuals*<br>Self-efficacy | Poor Quality<br>Case series with no comparator to control<br>Single site health system |

(*Continued*)

**Table 4.** (Continued)

| Reference | Population and setting | Implementation strategies | Absolute Difference % | Framework mapping | Study Quality and design |
|---|---|---|---|---|---|
| Kentwell [33] 2017 Australia | *Healthcare Professionals*: Gynaecology oncologist Specialist nurse Medical oncologist Genetic Counsellor *Patients*: All women with EOC *Healthcare Institution*: Publicly funded cancer unit at a major treating centre | Education for healthcare professionals GC embed in oncology services GC at MDT or documentation of GC and GT outcomes Genetics led referral pathway and triage | Outcome 1: GC referral +30.4% (95% CI 20.2–40.6, p≤0.001) Outcome 2. Time to gain access to GC and results GC referral I:2014–15–42 days 2015-16- 54.5 days GC referral to results 2014–15–106 days 2015-16- 140.5 days C: NR p value not provided Outcome 3. Patients identified with *BRCA* mutations I: 2014–2015 7/34; 20.6% 2015–2016 4/30; 13.3% C: NR p value not provided Outcome 4. Familial predictive GT uptake I:31/120 (28) C:NR p value not provided | *Implementation*: Acceptability • Satisfaction with mainstreaming *Service*: Efficiency • Time to gain access to GT and results Effectiveness • GC referral • Patients with identified gene mutations Equity • GT access • GC referral *Client*: Cancer prevention • Identification of hereditary Cancer **CFIR** *Inner setting* Readiness for implementation -access to knowledge and information - available resources | Poor Quality Case series with no control Single site health system |
| Tutty [48] 2019 Australia | *Healthcare Professionals*: Genetic counsellors Geneticist Gynaecology oncologist *Patients*: Women with EOC *Healthcare Institution*: Urban Australian Familial Cancer Centre | Genetic counsellor led telephone GC service for oncology services Genetics lead referral pathway and triage | Outcome 1. GC referral I: 284 C: NR p value not provided 2. GC and GT completion I: 284 C: NR p value not provided Outcome 3. Patients identified with *BRCA* mutations I: 26/284; 9% 12/284; 4% variants of unknown significance (VUS) C: NR p value not provided | *Implementation*: Acceptability • Satisfaction with TGC intervention Cost • Implementation cost *Service*: Efficiency • Cost of Resources to implement the intervention Effectiveness • GC referral and completion rate • GT completion • Patients with identified gene mutations Equity • GT access • GC referral Patient centeredness • Patients satisfaction with TGC intervention **CFIR** *Intervention Characteristics* • Cost *Outer setting* Needs & Resources of Those Served by the Organization *Process* Engaging • key stakeholders | Poor Quality Case series with no comparator to control Single site health system |

(*Continued*)

**Table 4.** (Continued)

| Reference | Population and setting | Implementation strategies | Absolute Difference % | Framework mapping | Study Quality and design |
|---|---|---|---|---|---|
| Bednar [35] 2017 USA | *Healthcare Professionals*: Physicians Genetic counsellors Advanced practice providers Nurses Clinical managers Physician trainees *Patients*: All women with EOC *Healthcare Institution*: An academic cancer centre's (regional and main campus clinics) | Education and direct access to GT via gynaecology Email notifications to refer EMR documentation and referral to GC Integrated genetic counsellor in oncology Scheduling GC appointments to co-inside with gynaecology | Outcome 1–3. GC referral I:561/1214 (46.2%) main campus clinic PCGT 84/151 (55.6%) regional clinic 653/1214 (53.8%) outside institution C: NR p value not provided I: AGCR 33/34 (97%) signed GC electronic referrals 14/72 (19.4%) email referrals C: NR p value not provided Outcome 4. GT completion I: 1214/1423 (85.3%) C: NR p value not provided Outcome 5. Patients identified with *BRCA* mutations I: 217/1214 (17.9%) C: NR p value not provided Outcome 6. Time to gain access to GC Absolute difference -119 days p value not provided | *Service*: Effectiveness • GT undertaken • GC referral • GC apt uptake Equity • GT access • GC referral • GT undertaken Timeliness • Time to GC apt *Client*: Cancer prevention • Identification of hereditary Cancer **CFIR** *Inner setting* Readiness for implementation • available resources *Process* Engaging • key stakeholders Executing | Poor Quality Case series with no comparator to control |
| Bednar [39] 2019 USA | *Healthcare Professionals*: Genetic counsellor Gynaecology oncologists Advanced practice registered nurses *Patients* Ovarian and uterine cancer patients *Healthcare Institution*: Regional hospital–single site with a gynaecology oncology clinic | Education for healthcare professionals Integrated GC in gynaecology EMR tracking and referral with e-mail notifications to refer | Outcome 1. GC referral I: 48/57 (84.2%) C: NR (p = 0.02) Outcome 2. GC and GT completion I: 43/48 (89.6%) completed GC 39/43(90.7%) completed GT C: NR (p = 0.03) Outcome 3. Patients identified with mutations I: 8/39 (20.5%) C: NR p value not provided | *Service*: Effectiveness • GT undertaken • GC referral • GC apt uptake • TT undertaken Equity • GT access • GC referrals • GT/TT undertaken *Client*: Cancer prevention • Identification of hereditary Cancer **CFIR** *Inner setting* Readiness for implementation • access to knowledge & information • available resources *Process* Engaging • key stakeholders Executing | Poor Quality Case series with no comparator to control Single site health system |

*(Continued)*

**Table 4.** (Continued)

| Reference | Population and setting | Implementation strategies | Absolute Difference % | Framework mapping | Study Quality and design |
|---|---|---|---|---|---|
| Percival [30] 2016 UK | *Healthcare Professionals*: Clinical nurse specialist in oncology Medical oncologists *Patients*: All women with EOC *Healthcare Institution*: Single centre urban hospital | Online education on pre-test GC for nurses Written information on *BRCA* testing for patients Written information for results significance Competency certificate after training complete for nurses Clinical Nurse specialist providing pre-test GC | Outcome 1. Patient satisfaction No difference in patient satisfaction between those consented by a nurse or a doctor No patients refused GT, or requested a GC appointment before GT. | *Implementation*: Acceptability • Satisfaction with mainstreaming intervention *Client*: Patients satisfaction with mainstreaming intervention **CFIR** *Inner setting* Readiness for implementation • access to knowledge and information *Outer setting* Needs & Resources of Those Served by the Organization *Characteristics of Individuals* Self-efficacy *Process* Engaging • key stakeholders | Poor quality Case series with no comparator to control Single site health system |
| Rahman [32] 2017 UK | *Healthcare Professionals*: Medical/clinical oncologists *Patients*: All women with EOC *Healthcare Institution*: Tertiary oncology centre | Education for healthcare professionals Testing protocol pathway Handouts for patients and healthcare professionals Standardised letters for results Standardised consent form | Outcome 1. GT completion I: 122/NR C: NR p value not provided Outcome 2. Patients identified with *BRCA* mutations I: 18/122 (14.8%) C: NR p value not provided Outcome 3. Time to gain access to GT, results & GC referral I: The time from sample receipt to result was between 14–48 working days—GC referral between 12–43 working days after MGT results -20/56 (36%) had MGT within 1 month of diagnosis C: NR No stats Outcome 4. Treatment management impact I: 11/18 (67%) no change in management 6/18 (33%) access PARP inhibitors C: NR No stats Outcome 5. Familial predictive GT uptake I: 11/ 15 family members of BRCA carriers having predictive GT C: NR No stats | *Service*: Effectiveness • GT undertaken *Equity* • GT access • GT undertaken Timeliness • Time to access GT, results and GC referral *Client*: Cancer prevention • Identification of hereditary Cancer **CFIR** *Inner setting* Readiness for implementation • access to knowledge and information *Process* Engaging • key stakeholders | Poor Quality Case series with no comparator to control Single site health system |

(*Continued*)

**Table 4.** (Continued)

| Reference | Population and setting | Implementation strategies | Absolute Difference % | Framework mapping | Study Quality and design |
|---|---|---|---|---|---|
| Plaskoinska [31] 2016 UK | *Healthcare Professionals*: Genetic Counsellor Oncologist Study co-ordinator *Patients*: All women with EOC *Healthcare Institution*: Rural and urban publicly funded hospitals of different sizes, ranging from smaller district general hospitals to large regional centres | Written information on pre-test GC for patients Genetics co-ordinated mainstreaming pathway Post–test GC by GC | Outcome 1. GT completion I: 232/281 (83%) C: NR p value not provided Outcome 2. Patients identified with *BRCA* mutations I: 18/232 (8%) C: NR p value not provided Outcome 3. Time to gain access to genetic test results I: Consent to results delivery 46 working days C: NR p value not provided | *Implementation*: Acceptability -Satisfaction with mainstreaming intervention Cost • Implementation cost *Service*: Effectiveness • GC referral • GT undertaken Equity • GT access • GC referrals • GT undertaken Efficiency -Time to gain access to GT results Patient centeredness -Patients satisfaction with mainstreaming intervention *Client*: Cancer prevention • Identification of hereditary Cancer **CFIR** *Intervention Characteristics* • Cost *Outer setting* Needs & Resources of Those Served by the Organization *Process* Engaging–key stakeholders | Poor Quality Case series with no comparator to control Single site health system |
| Cohen [47] 2016 USA | *Healthcare Professionals*: Medical Oncology Gastroenterology Surgery Pathology Laboratory Medical Genetics Genetic Counselling *Patients*: Patients with colorectal cancer *Healthcare Institution*: An outpatient cancer care centre for oncology patients treated at a tertiary academic National Cancer Institute (NCI)-designated Comprehensive Cancer Consortium | Triggered GC referral after abnormal IHC and MSI Handouts on referral process for LS for healthcare professionals Results tracking by nurse Shared GC e-mail to review all abnormal MSI and IHC Electronic communication with doctor Scheduling GC and CRC clinic appointments synchronously | Outcome 1. GC referral +9.4% (95% CI -7.9–26.8) p value not provided Outcome 2. Completion of GC +9.4% (95% CI -7.9–26.8) p value not provided Outcome 3. GT completion +10% (95% CI -47.6–67.6) p value not provided | *Service*: Effectiveness • GT undertaken • GC apt uptake Equity • GT access • GT undertaken **CFIR** *Inner setting* Readiness for implementation • available resources *Process* Engaging • key stakeholders | Poor Quality Cohort study with historical control Single site health system with no analysis on confounding variables or regression analysis on the characteristics inherent in the control verses the intervention population or health system |

*(Continued)*

**Table 4.** (Continued)

| Reference | Population and setting | Implementation strategies | Absolute Difference % | Framework mapping | Study Quality and design |
|---|---|---|---|---|---|
| Kemp [40] 2019 UK | *Healthcare Professionals*: All gynaecology oncology and cancer genetics health professionals unspecified *Patients*: Breast cancer patients *Healthcare Institution*: Publicly funded cancer unit at a major treating centre–cancer genetics services available | Education for healthcare professionals Testing protocol pathway Handouts for patients and healthcare professionals Standardised letters for results Standardised consent form | Outcome 1: GT completion I: 1184/1184 (100%) C: NR p value not provided Outcome 2. GC completion after GT I: 115/117 (98.3%) C:NR p value not provided Outcome 3. Patients identified with *BRCA* mutations I: 117/1184 (9.9%) C: NR p value not provided | *Implementation*: *Acceptability* • Satisfaction with mainstreaming intervention *Service*: *Effectiveness* • GT completion • Patients with identified gene mutations *Patient centeredness* • Patients satisfaction with mainstreaming intervention *Equity* • GT access • GT undertaken *Client*: *Cancer prevention* • Identification of hereditary Cancer **CFIR** *Intervention Characteristics* • Cost *Inner setting* Readiness for implementation • access to knowledge & information *Outer setting* Needs & Resources of Those Served by the Organization *Characteristics of Individuals* Self-efficacy *Process* Engaging | Poor Quality Case series with no comparator to control Single site health system |

*(Continued)*

**Table 4.** (Continued)

| Reference | Population and setting | Implementation strategies | Absolute Difference % | Framework mapping | Study Quality and design |
|---|---|---|---|---|---|
| Richardson l [54] 2020 Canada | *Healthcare Professionals*: Oncologists Genetic counsellor *Patients*: Breast and ovarian cancer patients *Healthcare Institution*: Population state based cancer program in Canada | Oncologist led pathway and communication Education for healthcare professionals Written information for clinician use Standardised consent form | Outcome 1. Acceptability **I**: Patients indicated comfort and acceptability with the GT process—no difference between oncology clinic-based model (OCB) and the traditional model (TM). OCB M = 4.54, SD = 0.71 vs TM M = 4.52, SD = 0.69. See Table 5 below C: NR Outcome 2. GC completed +58.6% (95% CI 49–68) and +8.5% (95% CI -8.2–25) in person and videoconference P< 0.001 OCB vs TM Outcome 3. GT completed +8.5% (95% CI -8.2–25 and +7.6% (95% CI -9.4–25, p = 0.015) OCB vs TM Outcome 4. Patients identified with *BRCA* mutations +3.1% (95% CI -6.7–13) p = 0.507 OCB vs TM Outcome 5. Time to gain access to GT results -212 days P< 0.001 OCB vs TM | *Implementation*: Acceptability • Satisfaction with mainstreaming intervention *Service*: Effectiveness • GT undertaken • GC referral Equity • GT access • GC referral • GT undertaken *Client*: Knowledge Acceptability Satisfaction Cancer prevention • Identification of hereditary Cancer **CFIR** *Inner setting* Readiness for implementation • access to knowledge & information • available resources *Outer setting* Needs & Resources of Those Served by the Organization *Process* Engaging • key stakeholders Reflecting & Evaluating | Good to Fair quality Cohort study with concurrent control State-wide health system with analysis on confounding variables or regression analysis on the characteristics inherent in the control verses the intervention population or health system Representation of patient population selective–all patients didn't complete survey. Small proportion of all patients included |
| Grinedal [53] 2020 Norway | *Healthcare Professionals*: Medical oncologist General Surgeon Gynaecologist Genetic Counsellor Geneticist *Patients*: Breast cancer patients *Healthcare Institution*: Regional and urban hospital in Norway | Education for healthcare professionals Testing pathway Written information for clinician use Standardised consent form | Outcome 1. GC referral I:131/356 (36.8%) C: NR p value not provided Outcome 2. GC completion I:125/356 (34.6%) C: NR p value not provided Outcome 3. GT completion I:125/131 (95.4%) C: NR p value not provided | *Service*: Effectiveness • GC referral • GC completion • GT completion *Client*: Equity • GT access • GC referral • GT undertaken **CFIR** *Inner setting* Readiness for implementation • access to knowledge & information • available resources *Process* Engaging • key stakeholders | Poor Quality Case series with no comparator to control |

(*Continued*)

**Table 4.** (Continued)

| Reference | Population and setting | Implementation strategies | Absolute Difference % | Framework mapping | Study Quality and design |
|---|---|---|---|---|---|
| Rumford [52] 2020 UK | *Healthcare Professionals*: All gynaecology oncology health professionals unspecified *Patients*: EOC patients *Healthcare Institution*: Publicly funded cancer unit at a major treating centre | Education for healthcare professionals Testing protocol pathway Handouts for patients and healthcare professionals Standardised letters for results Standardised consent form | Outcome 1. GC referral I:255/268 (95%) C: NR p value not provided Outcome 2. GC and GT completion I:255/268 (95%) C: NR p value not provided Outcome 3. Patients identified with *BRCA* mutations I:34/255 (13.3%) C: NR p value not provided Outcome 4: Time to gain access to GT I: Turnaround time between blood sample and return of GT result was 20.6 (11–42) calendar days C: Turnaround time of 148.2 calendar days prior to I Outcome 5. Treatment management impact I: 9/34 received a PARPi 5/34 receiving platinum-based chemotherapy–clinician intent to initiate PARPi chemotherapy 15/34 still receiving first-line (adjuvant) treatment or in remission —not eligible for PARPi 5/34 ineligible to receive PARPi C: NR p value not provided | *Service*: Efficiency • Time to gain access to GT *Effectiveness* • GC referral • GC completion • GT completion *Client*: Equity • GT access • GC referral • GT undertaken Cancer prevention • Identification of hereditary Cancer **CFIR** *Inner setting* Readiness for implementation • access to knowledge & information • available resources *Process* Engaging • key stakeholders | Poor Quality Case series with no comparator to control Single site health system |
| McLeavy [51] 2020 UK | *Healthcare Professionals*: Oncologist *Patients*: All EOC patients *Healthcare Institution*: Publicly funded tertiary referral centre | Education for healthcare professionals Testing protocol pathway Handouts for patients and healthcare professionals Standardised letters for results Standardised consent form | Outcome 1. Acceptability I: Decision Regret Scale 9.14±12.397– 14/29 (48.3%), reported no decision regret 26/29 (89.6%) were satisfied with their decision to pursue GT Participants produced relatively low MICRA scores regardless of mutation status C: NR p value not provided Outcome 2. GC completion I:170/170 (100%) C: NR p value not provided Outcome 3. GT completion I:170/170 (100%) C: NR p value not provided Outcome 4. Patients identified with *BRCA* mutations I:23/170 (13.5%) C:NR p value not provided | *Implementation*: *Acceptability* • Satisfaction with decision to undergo GT *Service*: *Effectiveness* • GT completed • Patients with identified gene mutations *Patient centeredness* • Patients satisfaction with mainstreaming intervention *Equity* • GT access • GT undertaken *Client*: *Cancer prevention* • Identification of hereditary Cancer **CFIR** *Inner setting* Readiness for implementation • access to knowledge & information *Outer setting* Needs & Resources of Those Served by the Organization *Process* Engaging • key stakeholders | Poor Quality Case series with no comparator to control Single site tertiary hospital setting |
| **Group 4: Results trend towards the control** | | | | | |

(Continued)

**Table 4.** (Continued)

| Reference | Population and setting | Implementation strategies | Absolute Difference % | Framework mapping | Study Quality and design |
|---|---|---|---|---|---|
| Long [42] 2018 Australia | *Healthcare Professionals*: Medical oncologist Surgeons Pathologist Genetic Counsellor and Geneticist Radiation oncologist Oncology nurses Oncology and genetics admin Palliative care *Patients*: Patients with colorectal cancer *Healthcare Institution*: NR | Education Standardised text for pathology reports and interpretation handouts Handouts on referral process for LS for healthcare professionals EMR documentation of GC/GT and/or referral via email MDT documentation of GC and pathology outcomes Results tracking | Outcome 1. Eligible for referral to GC Hospital A +7.24% (95% CI -2.3–17) Hospital B -1.88% (95% CI -9.4–5.6) Outcome 2. GC referral Hospital A -25% (95% CI -71-20) Hospital B +0.76% (95% CI -22-24) | *Service*: *Effectiveness* • GC referral **CFIR** *Inner setting* Readiness for implementation • access to knowledge & information • available resources *Process* Engaging • key stakeholders Reflecting and evaluating | Poor Quality Cohort study with historical control Two hospital sites but with no analysis on confounding variables or regression analysis on the characteristics inherent in the control verses the intervention population or health system |
| Swanson [37] 2018 USA | *Healthcare Professionals*: Surgeon Allied health staff Nurse Administrative Resident and fellow, Medical oncologist Geneticist Genetic counsellors *Patients*: All women with EOC *Healthcare Institution*: A tertiary care centre | Education for patients and healthcare professionals Family history collection proforma Handouts for patients EMR documentation of GC/GT and/or referral Testing protocol pathway Scheduling GC appointments directly at gynaecology clinic | Outcome 1. GC referral +27.4% (95% CI 11.1–43.7, p = 0.02) Outcome 2. GC completion -27.8% (95% CI -46.7 to -9.1) p value not provided Outcome 3. GT completion +20.6% (95% CI 5.9–35.4) p value not provided Outcome 4. Patients identified with *BRCA* mutations - 17.9% (95% CI– 40.9–5.1, p = 0.17) | *Service*: Effectiveness • GC referral • GC and GT completion Equity • GT access • GC referral • GT undertaken *Client*: Cancer prevention • Identification of hereditary Cancer **CFIR** *Inner setting* Readiness for implementation • access to knowledge and information *Process* Engaging • key stakeholders Executing | Fair Quality Cohort study with historical control Single site health system and no analysis on confounding variables or regression analysis on the characteristics inherent in the control verses the intervention population or health system |

MDT multidisciplinary team, EMR electronic medical record, EHR electronic health record GC Genetic Counselling, GT Genetic testing, I intervention, C comparator, NR not recorded, TT tumour testing, UTS universal tumour screening, MSI microsatellite instability testing, IHC immunohistochemistry, TNBC triple negative breast cancer, BrCa breast cancer, CRC colorectal cancer, VUS variant of unknown significance, EOC epithelial ovarian cancer, LS Lynch syndrome, PARPi poly (ADP-ribose) polymerase inhibitor

A documentation (referral guidelines and checklist in EMR GC referral) and system (GC EMR referral) intervention study [46] did not appear to have an impact on completion of GC (+9.9 (95% CI– 0.41–20.4) p = 0.505), but had a significant effect on GC referral (+12.7 (95% CI -0.04–25.4) p = 0.053). Regression analysis showed the intervention (p = 0.009), hereditary risk of cancer (p < 0.0001), and patients living in the metropolitan zone (p = 0.006) affected GC referral rates between the intervention and control [46] (Table 4). Three of the above studies [36, 37, 45] were not controlled for confounding variables or regression analysis on the characteristics inherent in the control verses the intervention population or health system apart from one [46] and the above interpretation of casual intervention impact needs to be interpreted with caution.

**Complex interventions with interdisciplinary practice.** Three studies [34, 38, 55] included an interdisciplinary practice complex intervention using a genetic counsellor or oncologist. One study used an interdisciplinary practice (GC embedded into oncology),

documentation (EMR GC and GT referral and completion) and system (GC appointment scheduling in oncology) complex intervention and led to a significant difference in GC referral (+22.8 (95% CI 16.7–29.4) p<0.00001) and GC completion rate (+45.5 (95% CI 33.6–57.6) p<0.00001) between the intervention and control [34]. Similarly, GT completion rate was impacted using an intervention consisting of education (oncology and breast health professionals' education on guidelines), interdisciplinary practice (increase in volume of GC and telemedicine consults) and documentation (referral to GC was made a standard of practice) [38]. A significant difference in GT completion was found by cancer or histology type for EOC (+29% (95% CI 16.8–41.2) p<0.05), TNBC < 60 yrs (+26.6% (95% CI 14.9–38.4) p<0.05) or breast cancer < 45 yrs (+15.7% (95% CI -7.5–6.1) p<0.05) between the intervention and control [38] (Table 4). An oncologist led GT intervention [55] with multidisciplinary team (MDT) communication and case management led to significant increase in GC referral (+25.4% (95% CI 16.4–34.3) p < 0.0001) and cancer prevention management (+22% (95% CI -16.2–60.3) p = 0.03), with less of an effect on GT completion between the intervention and control [55]. Similarly, a genetic counsellor at the MDT led to a significant difference in GC referral rates (+25% (95% CI 13.6–36.4) (P < 0.0001) p < 0.0001), between the intervention and control [43] (Table 4).

Among the complex interventions described above the common components of education [45, 36] and use of EMR to document and ensure GC referral occurred [34, 36, 37, 46] appear to have potential effects on outcomes such as GC referral, completion and GT completion.

**Complex interventions with no comparator.**  For the remaining seven studies, the composition of the interventions varied with five studies sharing a common complex intervention [29, 32, 40, 51, 52] in the UK and two studies from the USA [35, 39] and four studies with independent interventions [31, 48, 53, 54]. The complex interventions contained education [29, 32, 40, 35, 39, 51, 52], systems [35, 39], documentation [29, 32, 40, 35, 51–54] and interdisciplinary practice [35, 48] components (S4 Table). The potential intervention effect in relation to GT and GC completion rates were unclear with no comparator present to quantify an effect.

## Enhancing access to genetic counselling and genetic testing after universal tumour screening for colorectal and endometrial cancer

Five studies [41, 42, 44, 47, 39] described interventions aimed at enhancing access to GC and GT after UTS in colorectal and endometrial cancer (S4 Table). The interventions were varied with education, documentation, interdisciplinary practice or systems related domains (Table 1). The studies outcomes mapped to Proctor's evaluative framework [21] at the service or client level measuring potential effectiveness through GC referral [39, 41, 42, 44, 47] or GC [39, 41, 44, 47], GT completion rate [39, 41, 44, 47] or identification of hereditary cancer [39, 41, 44] and timely access to GC [44] (Table 2). CFIR [22] process and inner setting implementation factors were mainly addressed through engaging with health professionals in education or implementing the intervention [39, 41, 42, 44, 47] and two studies executing the implementation according to a plan [39, 42].

Two of the five studies showed a potential positive effect in favour of the intervention due to enhanced GC referral [44], completion of GC and GT [41, 44] and more patients being identified with hereditary cancer [41, 44] (Table 4).

**Complex interventions—Documentation and electronic system.**  Two studies [47, 41] used complex interventions, one consisting of education (handouts on LS referral process for clinicians) and systems (triggered GC referral after abnormal IHC and MSI, shared GC e-mail to review all abnormal MSI and IHC, electronic communication with physician, scheduling GC and CRC clinic appointments synchronously and results tracking by nurse) led to an

absolute difference in GC referral (+9.4 (95% CI -7.9–26.8) and GC (+9.4 (95% CI -7.9–26.8) or GT completion (+10 (95% CI -47.6–67.6) but with no statistical significant difference shown [47] (Table 4). Similarly, a systems (triggered GC referral after abnormal IHC and MSI, pathology communication via e-mail to surgeon) and interdisciplinary practice (patient navigators to ensure follow through to GC for abnormal IHC and MSI) led to an absolute difference in GC (+45.8% (95% CI 13.6–78.1) p = 0.020) and GT (+12.9%, (95% CI -24.7–50.4) completion and the identification of hereditary cancer (+28.8% (95% CI -21.5–79.2) between the intervention and control, with no statistical difference found [41] (Table 4).

**Complex interventions with interdisciplinary practice.**   One study consisting of interdisciplinary practice (GC embedded to review and communication abnormal IHC to patients and facilitate referral) and systems (triggered EMR GC referral after abnormal IHC and MSI to surgeon and documentation in EMR) complex intervention led to a significant difference in, GC referral (+44.7 (95% CI 28.1–60.5) p<0.001) and GC (+39.8% (95% CI 20.9–58.8) p<0.001) and GT (+39.8% (95% CI 21.1–58.5) p<0.001) completion rates between one arm of the intervention (genetic counsellor facilitation) and the control [44] (Table 4).

Among the complex interventions described above the common components of triggering GC referral after abnormal IHC results [41, 44, 47] and use of e-mail communication and review of IHC results between GC, pathology and surgeon [41, 47] appear to have potential effects on outcomes such as GC referral, completion and GT completion.

**Complex interventions with no comparator.**   One study [39] had an unclear intervention effect in relation to GC referral, GT and tumour testing completion rate and identification of hereditary cancer [39], as no comparator was available for assessment (S4 Table). Of note, all studies described under this theme had small sample sizes and none were controlled for confounding variables or regression analysis on the characteristics inherent in the control verses the intervention population or health system. As such the above interpretation of casual intervention impact on outcomes measured should be interpreted with caution.

**Efficiency and treatment management.**   Seven studies in breast and ovarian cancer measured the time taken to access GC or GT [32, 35, 52] and to receive the results of GT [29, 31, 32, 33, 54] after the intervention was implemented (Table 3). Six studies indicated efficiency in gaining access to GC and results [29, 31–33, 52, 54] and one study noted a reduction in time to access GC [35] (Table 5). The complex intervention in four studies representing single site hospitals with either GC services available on site [29, 35] or off site [32, 33] and one multiple centre study with regional and urban sites had unclear GC access for each site [31]. Three studies [44, 45, 54] with a historical or concurrent comparator in ovarian [45, 54], and colorectal [44] cancer showed a potential effect of the intervention in the reduction in time to gain access to GC and enhancement of familial GT uptake.

**System level outcome—Time efficiency.**   Two studies showed a reduction in time to receipt of GC [33, 35]. One study with a reduction of time within 42 and 54.5 days to GC and referral to results access within 106 and 140.5 days in two respective intervention time periods [33]. The other study reduced time to GC from 197 to 78 days when comparing the intervention and baseline times [35] and a fourfold reduction in time from GC to result was achieved in another study [29]. The time from sample receipt to result was reduced from 48 to 14 [32], 148.2 to 20.6 [52] days and post-test GC referral between 43 to 12 days [32]. Of note, sites with GC services available [29, 35] did not appear to show an advantage in time to gain access to GC in single site centres. However, the above studies did not compare the reduction in time to a comparator and firm conclusion cannot be drawn from the data presented.

Two studies with a historical or concurrent comparator [44, 54]—in the context of colorectal and ovarian cancer—showed a statistical difference with a reduction in time to gain access to GC. One study found a time reduction of 413 days (p<0.001) between intervention (when a

**Table 5. Implementation level outcomes of complex interventions in ovarian and subsets of breast cancer.**

| Study | Design | Acceptability | Cost |
|---|---|---|---|
| George [29] 2016 UK | Case series | Satisfaction and comfort with mainstreaming intervention | 13-fold reduction in genetics appointments with annual cost saving of 2.6 million |
| | Quantitative | I:105/105 patients were pleased to have had the genetic test | |
| | Survey | 15/15 clinicians were comfortable with consenting for genetic testing | |
| | Patients and Health professionals | C: NR | |
| Percival [30] 2016 UK | Case series | I: 108/300 Nurse | NR |
| | Quantitative | C: 192/300 Doctor | |
| | Survey | No difference in patient satisfaction between those consented by a nurse or a doctor | |
| | Patients and Health professionals | I: 75/108 patients consented by nurses completed a questionnaire. | |
| | | No patients refused GT, or requested a GC appointment before GT. | |
| | | C: NR | |
| | | Nurses satisfaction with pre-test GC training and role I: 5/6 nurses found the *BRCA* training helpful and saw *BRCA* testing was part of their role and felt supported. | |
| | | C: NR | |
| Plaskoinska [31] 2016 UK | Case series | I: 173/232 (75%) | I: £121 229 mainstreaming pathway |
| | Quantitative | low psychological impact to GT compared to cancer diagnosis (p<0.001). | C: £130 102 current standard pathway |
| | Survey | C: NR | Absolute difference = £8,873 |
| | Patients | I: 174/232 (75%) had enough information and time to decide to have GT | |
| | | C: NR | |
| Kentwell [33] 2017 Australia | Case series | A high level of comfort with; the process of consenting and delivering results | NR |
| | Quantitative | Medical oncologists (n = 6), | |
| | Health professionals | Less comfort in gynaecology oncologists and trainees (n = 5) | |
| Tutty [48] 2019 UK | Case series | I:97.2% and 94.3% were satisfied with the timing of the telephone call and information provided (n = 107) | I: $91.52 per woman tested (n = 72) |
| | Quantitative | C: NR | C: $ 107. 37 SIGC (n = 52) |
| | Survey | Low score for decisional regret (M = 4.25) | Absolute difference cost-saving—$15.85 |
| | Patients | 72% of the women indicating they had no regret regarding TFGT | |
| | | The psychological impact of receiving *BRCA*1/2 results was low | |
| | | (M = 7.9, SD = 7.5 for a negative test result; | |
| | | M = 16.8, SD = 9.7 for a positive test result; | |
| | | M = 12.0, SD = .6.3 for a VUS result) | |
| Kemp [40] 2019 UK | Case series | I: 129/259 patients surveyed | I: 2,500 genetics appointments |
| | Quantitative | 128/128 (100%) -pleased to have GT 124/129 | C: 50,000 genetics appointments |
| | Survey | 96.1% -happy that GT was via cancer team. | 95% reduction in genetic consultation |
| | Patients | 23/23 (100%) of cancer team members reported feeling confident to do *BRCA* testing during their consultation and believed that the process worked well | 85% reduction in time to test result |
| | | | Discounted QALY of 2746 compared to no testing |
| McLeavy [51] 2020 UK | Case series | I: **Decision regret scale** M 9.14±12.397–14/29 (48.3%), reported no decision regret | NR |
| | Quantitative | 26/29 (89.6%) were satisfied with their decision to pursue GT. Zero participants expressed clear dissatisfaction. | |
| | Survey | All participants felt sufficient time had been given to consider the offer of mainstreamed genetic testing. | |
| | Patients | **Participants produced low MICRA scores** | |
| | | Distress M = 2.66 ± SD 4.108 | |
| | | Uncertainty M = 5.07± SD 4.154 | |
| | | Positive experiences M = 3.36± SD 4.093 | |
| | | Familial risk M = 7.05± SD 3.027 | |
| | | Ability to cope M = 0.26± SD 0.656 (coping harder with MGT) | |
| | | M = 2.46± SD 2.134 (coping easier with MGT) | |
| | | 26/29 (89.6%) felt adequately supported by the oncology department. | |

(*Continued*)

**Table 5.** (Continued)

| Study | Design | Acceptability | Cost |
|---|---|---|---|
| Richardson [54] 2020 Canada | Concurrent cohort | 259/400 completed survey– 57/259 from the oncology clinic based (OCB) and 202/259 from the traditional model (TM) | NR |
| | Quantitative | **Patient Acceptability Scale** | |
| | Survey | OCB M = 4.54, SD = 0.71 vs TM M = 4.52, SD = 0.69 | |
| | Patients | 8/19 oncologists completed survey– 5/8 strongly agreed or agreed with 'the process for carrying out multi-gene panel testing worked well', | |
| | Healthcare professionals | **MICRA score**–Distress | |
| | | OCB M = 4.53, SD = 5.65 vs TM M = 3.37, SD = 5.24 | |
| | | Uncertainty | |
| | | OCB M = 9.51, SD = 8.19 vs TM M = 10.02, SD = 6.88 | |
| | | Positive experience | |
| | | OCB M = 6.00, SD = 5.78 vs TM M = 4.45, SD = 4.66 | |
| | | **Decisional conflict scale** | |
| | | Uncertainty | |
| | | OCB M = 22.57, SD = 19.52 vs TM M = 23.36, SD = 21.25 | |
| | | Informed | |
| | | OCB M = 19.71, SD = 14.04 vs TM M = 18.04, SD = 17.38 | |
| | | Values Clarity | |
| | | OCB M = 24.13, SD = 17.04 vs TM M = 24.22, SD = 19.73 | |
| | | Support | |
| | | OCB M = 25.18, SD = 18.23 vs TM M = 26.61, SD = 20.94 | |
| | | Effective Decision | |
| | | OCB M = 13.16, SD = 14.32 vs TM M = 15.21, SD = 19.43 | |
| | | **Genetic Counselling Outcome Scale** | |
| | | OCB M = 120.17, SD = 16.78 vs TM M = 120.93, SD = 15.15 | |
| Shipman [50] 2017 UK | Qualitative | **Motivations and Influences re Offers of GT** | NR |
| | Interviews | Genetic Testing was Just Not Disruptive in the Context of Cancer Diagnosis | |
| | 17 Patients and Health Professionals | Illustrative Quote "I mean I was going- I was going through chemo at the time an, you know, I just wanted to get through the chemo (laughing tone) I really didn't really care about you know, as long as I was gonna be all right, that was all I was concerned about . . .And that's made a big difference to my attitude to all the tests and studies and everything" (EOC patient with mutation identified) | |
| | | **Staff Anxieties** | |
| | | "Once they've had a diagnosis they're bamboozled with the idea of all the treatment options in front of them or they might be post-surgical and facing chemo. . .and they're probably not at the most receptive point to consider this. They're already on this sort of rollercoaster, they're in shock" (Research staff) | |
| Meiser [49] 2012 Australia | Qualitative | **Acceptance of TFGT** | NR |
| | | "It's the same as having an operation. It's not very pleasant but if you have to have it, you have to have it" (Invasive ovarian cancer patient eligible for BRCA testing) | |
| | Interviews | **Perceived advantages of TFGT** | |
| | | "But I just think that more information, yes it's scary, but the more you know the better off you are to be able to make a decision" (Invasive ovarian cancer patient eligible for *BRCA* testing) | |
| | 22 Patients | **Perceived need to make TFGT a routine test** | |
| | | "I believe that it should be incorporated into the overall testing because then it ultimately gives the treating oncologist like a much bigger picture and the full picture" (Invasive ovarian cancer patient eligible for *BRCA* testing) | |

GC Genetic Counselling, GT Genetic testing, I intervention, C comparator, NR not recorded, TFGT treatment focused genetic testing, OCB oncology clinic based, TM traditional model, MICRA Multidimensional Impact of Cancer Risk Assessment

GC was involved in receiving IHC results along with the surgeon and facilitated results communication between patient and making a GC referral) and control (no GC involved and all IHC results sent to surgeon and GC referral made at surgeon discretion) [44]. The other study showed a time reduction to gain access to GT results of 212 days (p<0.001) between the intervention (direct access to pre-test GC and panel GT through oncologists in an oncology clinic-based model with post-test GC provided by a genetic counsellor) versus the control (referral to GC) [54] (Table 4 and S4 Table).

**Client level outcome—Treatment management.** Four studies described the treatment impact of direct access to GT in routine oncology care for EOC [29, 33, 52, 55] (Table 4 and S4 Table). Treatment was informed in 132/207 of ovarian cancer patients either at first line therapy or relapse of their disease with 20/23 women [29] and 6/18 women with *BRCA* mutations gaining access to PARPi [32]. Ovarian cancer patients with BRCA pathogenic variants (32/33) had breast cancer surveillance [29] and breast cancer patients had significantly more risk reducing measures compared with the control (+22% (95% CI -16.2–60.3) p = 0.03) [55] with the oncologist led intervention (oncologist led pathway, communication and MDT) compared to control (usual care and referral pathway to a genetics unit). PARPi was received by 9/34 *BRCA* pathogenic variant women with ovarian cancer, with 5/34 indicated to initiate PARPi, in the future [52] (Table 4 and S4 Table).

Three studies described the family management of *BRCA* through the uptake of predictive GT in family members [32, 33, 45]. At risk family members accessed predictive GT with varying degrees from 31/120 [33] to 11/ 15 [32] in the study time period. Predictive GT and identification of *BRCA* carriers significantly increased per histology subtype in the intervention (education on benefit of GC referral for cancer prevention and inclusion on pathology reporting) arm from 2.54 to 3.27 (p = 0.071) and 1.62 and 2.18 (p = 0.009) compared to the control (usual care with no education or pathology reporting GC recommendations), respectively [45] in ovarian cancer families (Table 4 and S4 Table).

**Implementation level outcome and factors.** Eight studies assessed the acceptability of the intervention used to introduce routine GT into oncology care of EOC [29–31, 33, 48, 51, 54] and for subsets of breast cancer patients [40, 54] (Tables 3 and 5). Four studies assessed satisfaction of the new process from the patients and healthcare professional perspective using survey style questions [29, 30, 40, 54] and mapped to CFIR outer setting of patient needs and resources. Implementation factors addressed through CFIR showed that only a minority of studies focused on reflecting and evaluation the implementation efforts through health professionals' feedback about the intervention [40, 42, 54], their belief with regards to their ability to undertake the intervention [29, 30, 33, 40] and the cost or cost savings with such efforts [29, 31, 40, 48] (Tables 3 and 5).

**Implementation level outcome—Acceptability.** Two studies [29, 40] showed most patients were pleased to have had the genetic test and via the cancer team [40] and all healthcare professionals involved were comfortable with consenting for GT [29] and felt confident to provide *BRCA* testing, believing the process worked well [40]. Patient satisfaction between GT consented by a nurse or a doctor was not impacted amongst surveyed participants [30], with no patients refusing GT or requesting a GC appointment beforehand with nurses consenting [30]. The majority of nurses found the *BRCA* training helpful, saw *BRCA* testing as part of their role and felt supported [30]. Five studies assessed satisfaction either from the patient [31, 48, 51, 54] or the healthcare professional perspective [33, 54]. The majority of patients were satisfied with the timing of the telephone call, their decision to pursue GT [51] and the information provided, indicating they had sufficient information and time to decide to have GT [48, 51]. Overall, women indicated high scores of satisfaction based on the Genetic Counselling Satisfaction Scale (GCSS [58]) [48, 54]. Healthcare professionals indicated a high level of

comfort with the process of consenting to and delivering results for GT amongst the medical oncologists [33, 54], but less so amongst the gynaecology oncologists and trainees [33]. All except two [48, 54] of the above studies used self-designed survey question with no validated measures of satisfaction to evaluate this aspect of acceptability. The reliability and validity of the results are limited in this regard.

Four studies evaluated the psychological impact of receiving GT during the cancer diagnosis and treatment period [31, 48, 51, 54]. Low patient scores on psychological impact to GT compared to their cancer diagnosis were found with validated measures such as the impact of events scale (IES) [59] and Depression, anxiety and stress scale (DASS -21 [60]) [31]. The only difference in population characteristics was younger age and mutation status, leading to more intrusive thoughts (IES intrusion r = −0.172, p = 0.026) and significantly more stress (DASS stress r = 0.162, p = 0.014) and cognitive avoidance scores based on *BRCA* pathogenic variant status, respectively [33]. Patients indicated a low score for decisional regret and psychological impact [48, 51, 54] of GT results, in relation to their decision to undergo treatment-focused genetic testing (TFGT) and receiving results [48]. Validated measures such as the decisional regret scale [61] and the multidimensional impact of risk assessment [62] were used respectively. Validated measures strengthen the results reporting acceptability of the new approach from the patients' perspective and can be reliably reproduced in future studies on acceptability of interventions.

Two qualitative studies support the acceptability of routine GT in oncology for ovarian cancer patients [49, 50] (Table 5). Both the 12 ovarian patients who had TFGT and those who were asked hypothetically found the concept of TFGT to be acceptable and wanted it as a routine test to inform their care [49]. Seventeen patients and five staff members offered or involved in TFGT [31] did not see GT as disruptive in the overall experience of having a cancer diagnosis [50]. However, some staff expressed concern about overburdening recently diagnosed patients undergoing chemotherapy. Both qualitative studies support the implementation outcome of acceptability of incorporating GT into routine oncology.

**Implementation level outcome—Cost.**   Four studies assessed the cost and resources needed to implement routine GT [29, 31, 40, 48] into oncology when mapped to both Proctor's evaluative lens and CFIR intervention characteristics of cost (Tables 3 and 5). Two studies [29, 40] evaluated the reduction in genetics appointments and cost. A 13-fold reduction in genetics appointments, with an annual cost saving of 2.6 million for the mainstreaming pathway in ovarian cancer was found in one study [29]. Another study found a 95% reduction in genetic consultation, with an 85% reduction in time to test result for the mainstreaming pathway for subsets of breast cancer [40]. The cost of the pathway or testing per patient in the mainstreaming versus the traditional pathway showed a cost reduction of UK£8,873 [31] for the former and AUS $15.85 [48] for the later. A robust economic analysis was not evident in the above studies and many lacked a comparator to strengthen the evidence regarding cost reduction.

## Discussion

This systematic review aimed to examine health system interventions used to increase the uptake of GC and GT in oncology services to identify hereditary breast, ovarian, colorectal and endometrial cancer. The evidence indicates that complex interventions have a potentially positive effect on GC and GT completion rates in oncology services. Twenty-five studies identified intervention characteristics, with eight of these also describing implementation factors that influenced access to GT, GC and identification of hereditary cancer. The health professional groups targeted by the interventions were varied. About one-quarter of studies included an implementation science model or framework to guide intervention design or implementation

and high priority CFIR constructs were not always applied in studies to understand the implementation factors.

Many types of health professionals were targeted with the designed intervention mainly focusing on oncologists or surgeons, advanced practice nurses or clinical nurse specialists, pathologists or genetic counsellors. A broader range of healthcare professionals such as radiographers, gastroenterologists and colorectal surgeons could be included in future GT integration programs in oncology through breast or CRC screening programs, once these health professionals have been appropriately skilled [63–65].

As described above, about one quarter of the studies included an implementation science framework or a quality improvement model to guide intervention design and implementation. However, the interaction between the intervention and the theory behind the strategies was not addressed from the above studies and thus limits generalizable lessons. Evidence based pre-implementation research underpinned with implementation theory is crucial in guiding the development and evaluation of interventions [66]. In future, more studies using a theory-based evaluation of implementation-level outcomes are needed to better understand intervention implementation efforts. Some high priority CFIR constructs identified by the Implementing GeNomics In PracTicE (IGNITE) [67] model were present in the included studies apart from intervention characteristics (relative advantage), individuals characteristics (knowledge and belief about the intervention) and inner setting (implementation climate). Future genomics implementation research incorporating all high priority constructs would aid in a broader understanding of genomics implementation factors in diverse contexts and systems.

## Strengths and limitations

Our review provides a comprehensive and rigorous assessment of interventions to integrate GT in oncology. It is the first to assess GT mainstreaming programs and apply a recognised implementation science outcome [21] and determinant framework [22] to identify common client, service or implementation outcomes in assessing intervention effectiveness and implementation factors. The data aligned with some of the high priority CFIR constructs identified as important to assess in genomics implementation research. Most studies used complex interventions that can inform implementation strategies for future genomics implementation research.

Regarding limitations, the inclusion of a variety of study designs and intervention types precluded the conduct of a meta-analysis. With this variation, no specific quality measure was available to evaluate the diverse literature. The adaptation of the NOS addressed the potential bias in studies by developing specific implementation intervention assessment criteria (S5 and S6 Tables). Many studies had poor methodological design and reported on few outcomes across the implementation level. A lack of focus on implementation outcomes impacts the understanding of what will contribute to the longer-term sustainability of GT integration in health systems.

Across the included studies, there was a lack of consistency in classification of interventions and the strategies used to implement them. A classification system (Table 1) was created to apply parameters to intervention characteristics and provide a means to analyse a potential effect. However, the mechanisms and data that explain why an intervention may, or may not, yield change were unclear. Those studies that did use a framework or theory rarely addressed the interaction between these and the strategies selected. This limits the generalizable lessons that might have been learned from these studies. Finally, the calculated absolute difference value cannot conclusively provide accurate estimates about the impact of a particular

component of an intervention due to the heterogeneity of intervention characteristics, variability of health systems and range of health professionals involved.

The majority of studies were single site, urban hospital settings, which limits generalisability. However, a wide variety of health systems structures implementing complex interventions were represented in the included studies, allowing understanding of the possible implementation strategies that may work in similar settings.

## Implications for research and policy

Our findings indicate that complex interventions–using systems, education, documentation and interdisciplinary practice–have a potential positive effect on GC and GT outcomes in various cancer types. The majority of the studies were of small sample size and did not collect longitudinal data or utilize an evaluation framework to underpin the intervention effectiveness findings with an assessment of all outcomes at the client, service and implementation level. Future research requires more rigorous study and evaluation designs by examining the patient, provider, organization and policy levels of healthcare to improve health outcomes [66].

The optimisation of patient outcomes requires implementation research to align with the real-world problems and priorities of healthcare organisations [2]. As more GT is mainstreamed into routine oncology care, future interventions need to fit with organizational workflows and processes to encourage successful implementation. Hybrid study designs allow for simultaneous measurement of intervention and implementation effectiveness [68]. Similarly, step wedge designs allow interventions to be introduced and evaluated in a staged way and to compare the effects of implementation [69] across different hospital settings [70].

Furthermore, pre-implementation research that identifies organisation characteristics and barriers and then tailor interventions to address these adds value to the likely adoption of new innovations [71]. In our review, a minority of studies assessed acceptability as part of the intervention implementation efforts; evidence from other qualitative studies supports the acceptability and feasibility of integrating GT into oncology services [72]. Future research with a pre-implementation focus on implementation outcomes and defined intervention characteristics could enhance understanding of the factors that influence GT integration implementation efforts.

## Conclusion

This systematic review contributes new knowledge to the genomics implementation field by summarizing and assessing the characteristics and outcome findings of mainstreaming GT programs and uptake of GT after UTS. The existing evidence on intervention effectiveness suggests GT mainstreaming programs increase access to GC and GT in oncology services. However, there is a significant gap in understanding the interaction between the intervention and implementation theory to harness generalizable implementation strategies. Future primary research studies with robust methodological quality informed by theory are required. Results from this systematic review could inform future implementation strategies to integrate genetics into routine care of oncology health systems.

## Supporting information

**S1 Table. Search strategy Medline, Embase, PsychINFO (ovid) up to 26.05.20\***.
(PDF)

**S2 Table. Search strategy CINAHL (EBSCO) up to 26.05.20\***.
(PDF)

**S3 Table. Inclusion and exclusion criteria for routine genetic testing integration intervention studies in oncology.**
(PDF)

**S4 Table. Complex interventions to increase genetic counselling, testing and identification of hereditary in ovarian, breast, colorectal and endometrial cancer.**
(PDF)

**S5 Table. Assessment of risk of bias of included cohort studies.**
(PDF)

**S6 Table. Assessment of risk of bias of included case series.**
(PDF)

**S1 File. References: Included studies.**
(PDF)

**S2 File. Definitions.**
(DOCX)

**S1 Checklist.**
(DOC)

## Acknowledgments

The authors are grateful to Suzanne Hughes and Chelsea Carle for supporting the development of the systematic review process.

## Author Contributions

**Conceptualization:** Rosie O'Shea, Natalie Taylor, Yoon Jung Kang, Nicole M. Rankin.

**Formal analysis:** Rosie O'Shea, Ashley Crook, Nicole M. Rankin.

**Investigation:** Rosie O'Shea, Ashley Crook, Chris Jacobs, Sarah Lewis.

**Methodology:** Rosie O'Shea, Natalie Taylor, Chris Jacobs, Nicole M. Rankin.

**Supervision:** Sarah Lewis, Nicole M. Rankin.

**Writing – original draft:** Rosie O'Shea.

**Writing – review & editing:** Rosie O'Shea, Natalie Taylor, Ashley Crook, Chris Jacobs, Yoon Jung Kang, Sarah Lewis, Nicole M. Rankin.

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
