## [Decision Letter · Decision Letter 0]

22 Feb 2021

PONE-D-20-35958

Health system interventions to integrate genetic testing in routine oncology services : a systematic review

PLOS ONE

Dear Dr. O'Shea,

Thank you for submitting your manuscript to PLOS ONE. After careful consideration, we feel that it has merit but does not fully meet PLOS ONE’s publication criteria as it currently stands. Therefore, we invite you to submit a revised version of the manuscript that addresses the points raised during the review process.

We look forward to receiving your revised manuscript.

Kind regards,

Alvaro Galli

Academic Editor

PLOS ONE

Journal Requirements:

2. We would suggest  that some of the  information  at present shown in the Supplementary materials should be included in the main text; for example, the main text should contain a  table reporting all the included studies and their main characteristics, and the results of the quality assessment.

Reviewers' comments:

Reviewer's Responses to Questions

**Comments to the Author**

1. Is the manuscript technically sound, and do the data support the conclusions?

Reviewer #1: Yes

Reviewer #2: Yes

2. Has the statistical analysis been performed appropriately and rigorously? 

Reviewer #1: N/A

Reviewer #2: Yes

3. Have the authors made all data underlying the findings in their manuscript fully available?

Reviewer #1: Yes

Reviewer #2: Yes

4. Is the manuscript presented in an intelligible fashion and written in standard English?

Reviewer #1: Yes

Reviewer #2: Yes

5. Review Comments to the Author

Reviewer #1: The authors present an extensive systematic review of interventions aiming to increase referrals to, and uptake of, genetic counseling and testing in oncology for breast, ovarian, and tumor screen-positive colon and uterine cancers. The summary tables represent an extraordinary amount of work, and there is clearly a high level of rigor in adhering to current standards for implementation science and reporting of systematic reviews.

My minor comments are as follows:

1) Line 69/70: There are already established clinical guidelines for GT for colon/uterine cancer; I don't think they are "emerging".

2) Line 88-89: The statement that "GT access is via referral to genetics services"; in many localities (particularly in the US) this is not the case; colorectal surgeons and oncologists and GYN oncologists have been ordering GT directly without referral to genetics clinics for many years.

3) Similarly for line 96/97: "GT is now being introduced..."-- GT has in fact been ongoing for nearly two decades; while there is quite a lot of heterogeneity in access and implementation and processes, I don't believe it is accurate to say that it is "now being introduced". At least in the US, the NCCN guidelines have had recommendations for routine testing of certain patient populations for many years.

4) Lines 133-161: It may be that the formatting and use of bullet points did not translate well into the manuscript, but the inclusion criteria should be stated more clearly and with more attention to grammar. It partly reads as a set of bullet point criteria, but starts out as though it will read as a sentence- this is confusing to the reader.

5) Lines 165-166: The sentence does not make sense-- I think you meant to say "Additionally, * a study was excluded* if the outcomes..... "

6) The results section overall is very comprehensive, but the text paragraphs summarizing each grouping is fairly dry, and does not really add much in the way of practical information to the reader. While the details are nicely presented in the tables, I think that readers who are looking to understand "which interventions have been most effective, and might they apply to my clinic" would like to see additional observations about specific interventions here. For example, the authors might note when a specific type of intervention (e.g. pathology report language; physician education) was a component of multiple studies showing an effect, or similar color; otherwise this section has little information to guide a reader who is a health care provider looking for evidence that a particular intervention might be effective. I appreciate that an implementation science audience would likely disagree with my suggestion though!

7) Similarly the use of p values in the majority of the results text is not as helpful as effect sizes; a p-value by itself is not informative and most journals are moving away from emphasizing p-values out of context.

8) I don't see a clear justification for why a positive genetic test result is a desired outcome- in practical terms, a completed test result has value whether it is positive or negative. For example, a woman with ovarian cancer who undergoes testing and does not have a BRCA1/2 mutation now knows that iPARP therapy is not likely to be useful. In studies of family members undergoing cascade testing, those who test negative for the familial mutation are released from high risk screening-- that is extremely valuable. While some studies certainly reported on mutation rates, the way this is presented in this manuscript implies that positive tests are a desirable outcome that "should be increased", in clinical practice this is not the case.

9) Line 467- should say "except" instead of "exempt"

10) Lines 567-571- Given the wide variety of practice settings and heterogeneity in oncology clinics around the world, I don't know that it is fair to imply that there are uncertainties about the "true effects of interventions...". While single site studies do limit generalizability, they are in fact very helpful for informing possible interventions that might be effective *at similar clinics*. For example, a clinic intervention that trains nurses to do pre-test consent and sees an increase in the proportion of ovarian cancer patients who complete genetic testing prior to starting treatment would be highly valuable for other clinics who do not have an on-site genetics clinic but do have nurses who have access to genetics training, while other clinics who have on-site "embedded" genetic counselors in their gyn clinic might not have a use for that particular intervention. In other words, heterogeneity is not necessarily a bad thing. I note that this may be more of a fact for US-based oncology clinics vs clinics in systems with universal health care where there is a more uniform approach.

Reviewer #2: The authors completed a SR on integrating genetic testing into routine oncology care. This SR covers a relevant research question and the authors report on key outcomes that are especially important for researchers trying to implement genomic programs. Although well written and the methods are well documented, a few minor comments are suggested below to help enhance the analysis.

Abstract:

• Line 43: what was the cutoff date? 2011-?

• Line 46: define complex intervention. Quantify “increase in access to genetic counseling and testing in routine oncology practice”. How much of an increase?

Background:

• Line 64: specify the type of genetic testing. Somatic vs germline testing.

• Line 111: provide an example of a single unit vs complex intervention

Methods

• Line 133: Although not required, it would be helpful to include timing/length of follow up as part of the inclusion/exclusion criteria.

• Line 193: double check how the tables/figures are numbered. Table 3 is mentioned here but Tables 1 and 2 were not mentioned prior to this.

• Line 201: how was the quality of RCTs assessed?

Results:

• Line 277: I like how the studies were mapped to outcomes and CFIR constructs. One challenge in implementing programs is knowing which outcomes and constructs to use and the optimal combination of them. These outcomes and constructs are reported individually, but what would be helpful from a practice standpoint is knowing the combination of constructs that studies implemented and seeing how that impacted outcomes. Perhaps the CFIR constructs could be included in Table 4?

• It’s a bit difficult to follow all the tables. Perhaps to simplify this, a diagram outlining how the studies were mapped to Proctor and CFIR could be diagrammed and mention of which table the results are in could be referenced in the figure. Just a thought to help the reader know which table to reference for specific information.

Tables:

• Is there a table where the study design is summarized? This would be helpful to know in tables such as Table 4.

• Table 1: I’m still confused about what a “complex intervention type” is. Is “Education” considered complex because there were 4 implementation strategies that were involved in a study? And what about the single units? Is there a list of these somewhere?

Supplemental materials:

• All titles for tables and figures should start with “Supplemental”.

6. PLOS authors have the option to publish the peer review history of their article (what does this mean?). If published, this will include your full peer review and any attached files.

Reviewer #1: **Yes: **Lisa Madlensky

Reviewer #2: No

---

## [Author Response · Author response to Decision Letter 0]

18 Mar 2021

Dr. Alvaro Galli

Academic Editor

PLOS ONE

19.03.21

Dear Dr. Galli,

PONE-D-20-35958, entitled "Health system interventions to integrate genetic testing in routine oncology services: a systematic review" 

We are grateful to the reviewers for taking the time to provide detailed and constructive feedback that has certainly improved the quality of our revised manuscript in both content and clarity. We thank the reviewers for acknowledging the rigor and reporting of the studies for our systematic review and the well written manuscript examining interventions and implementation factors for integrating genetic testing into routine oncology setting. 

We have highlighted the changes in the revised manuscript and re-submitted a clean version for your consideration. Our detailed responses are as follows:

Journal Requirements

1.COMMENT TO THE AUTHOR:

RESPONSE: The formatting of the title page has been revised as per the PLOS ONE template style above. See tracked changes on page 1. The heading style, table and figure and supporting information citations and captions have been updated as per required format. Reference list has been check to comply with Vancouver style referencing. Reference number 75 has been removed and replaced with the correct reference. See tracked changes throughout the manuscript.

2.COMMENT TO THE AUTHOR:

We would suggest that some of the information at present shown in the Supplementary materials should be included in the main text; for example, the main text should contain a table reporting all the included studies and their main characteristics, and the results of the quality assessment.

RESPONSE: The characteristics of the included studies including the study design characteristics are included in Table 2. Framework mapping and quality assessment including the study design have now been included in Table 4. Both of these points addressed as per reviewer 2 comment 4.

3.COMMENT TO THE AUTHOR:

We note that the grant information you provided in the ‘Funding Information’ and ‘Financial Disclosure’ sections do not match.

RESPONSE: Funding information has been corrected. There are no specific grant numbers to reference as the funding was through a scholarship award from both entities described and not specific grant funding.

4.COMMENT TO THE AUTHOR:

Please include captions for your Supporting Information files at the end of your manuscript, and update any in-text citations to match accordingly. Please see our Supporting Information guidelines for more information: http://journals.plos.org/plosone/s/supporting-information.

RESPONSE: Caption information has now been included for supporting information at the end of the manuscript and cited in the text as Table in S1 Table, as per requirements indicated in the above link. See tracked changes throughout for citing and at the end of manuscript page 110 of tracked changed manuscript for captions. 

REVIEWER 1

The authors present an extensive systematic review of interventions aiming to increase referrals to, and uptake of, genetic counseling and testing in oncology for breast, ovarian, and tumor screen-positive colon and uterine cancers. The summary tables represent an extraordinary amount of work, and there is clearly a high level of rigor in adhering to current standards for implementation science 

and reporting of systematic reviews.

My minor comments are as follows:

1. COMMENT TO THE AUTHOR:

1. Line 69/70: There are already established clinical guidelines for GT for colon/uterine cancer; I don't think they are "emerging".

RESPONSE: We agree with the reviewer that clinical guidelines are established for GT for colon/uterine cancer. We have edited lines 69-70 to read “Established clinical guidelines for directing access to GT for endometrial and colorectal cancers exist in the USA, UK and Australia”

2. COMMENT TO THE AUTHOR:

Line 88-89: The statement that "GT access is via referral to genetics services"; in many localities (particularly in the US) this is not the case; colorectal surgeons and oncologists and GYN oncologists have been ordering GT directly without referral to genetics clinics for many years.

RESPONSE: We agree with the reviewer that direct access to GT in the USA has been available through surgeons and oncologists for many years. We have edited lines 89-94 to indicated direct access to GT is available and that Australia has only recently introduced this testing policy change in 2020. See track changes lines 89-94.

3. COMMENT TO THE AUTHOR:

Similarly for line 96/97: "GT is now being introduced..."-- GT has in fact been ongoing for nearly two decades; while there is quite a lot of heterogeneity in access and implementation and processes, I don't believe it is accurate to say that it is "now being introduced". At least in the US, the NCCN guidelines have had recommendations for routine testing of certain patient populations for many years.

RESPONSE: We agree with the reviewer that recommendations for routine testing for certain cancers has been available in the US for many years. To reflect the difference in timeframes in Australia where direct access to GT for CRC and EC patient populations has just been introduced in 2020, we have edited lines 98-104 to indicated how learning from other jurisdictions mainstreaming strategies can inform CRC and EC mainstreaming in Australia. See track changes lines 98-104. 

4. COMMENT TO THE AUTHOR:

Lines 133-161: It may be that the formatting and use of bullet points did not translate well into the manuscript, but the inclusion criteria should be stated more clearly and with more attention to grammar. It partly reads as a set of bullet point criteria, but starts out as though it will read as a sentence- this is confusing to the reader.

RESPONSE:The formatting of the inclusion criteria section has been improved for clarity, grammar, sentence structure and the use of numbers and bullet points to stratify the criteria. See tracked changed manuscript lines 137-170.

5. COMMENT TO THE AUTHOR:

Lines 165-166: The sentence does not make sense-- I think you meant to say "Additionally, * a study was excluded* if the outcomes..... "

RESPONSE: We thank the reviewer for highlighting the need to clarify this sentence. We have edited as suggested above, see track changed manuscript lines 173-174.

6. COMMENT TO THE AUTHOR:

The results section overall is very comprehensive, but the text paragraphs summarizing each grouping is fairly dry, and does not really add much in the way of practical information to the reader. While the details are nicely presented in the tables, I think that readers who are looking to understand "which interventions have been most effective, and might they apply to my clinic" would like to see additional observations about specific interventions here. For example, the authors might note when a specific type of intervention (e.g. pathology report language; physician education) was a component of multiple studies showing an effect, or similar color; otherwise this section has little information to guide a reader who is a health care provider looking for evidence that a particular intervention might be effective. I appreciate that an implementation science audience would likely disagree with my suggestion though!

RESPONSE: The results section now includes more detailed description of the specific components of complex interventions that have shown a positive effect on outcomes. See tracked changed manuscript for detailed intervention component descriptions lines 344-358, 364-370,374-377,387-405. At the end of the results sections describing complex interventions lines 408-410 and 459-462 note when there is commonality in intervention components that showed a positive effect on outcomes. 

7. COMMENT TO THE AUTHOR:

Similarly the use of p values in the majority of the results text is not as helpful as effect sizes; a p-value by itself is not informative and most journals are moving away from emphasizing p-values out of context.

RESPONSE: The results section now includes effect sizes and confidence intervals along with p values to detail better the potential effects of complex intervention types on outcomes. See tracked changed manuscript lines 344-358, 364-370,374-377,387-405. This information is also in Table 4 and S4 Table

8. COMMENT TO THE AUTHOR:

I don't see a clear justification for why a positive genetic test result is a desired outcome- in practical terms, a completed test result has value whether it is positive or negative. For example, a woman with ovarian cancer who undergoes testing and does not have a BRCA1/2 mutation now knows that iPARP therapy is not likely to be useful. In studies of family members undergoing cascade testing, those who test negative for the familial mutation are released from high risk screening-- that is extremely valuable. While some studies certainly reported on mutation rates, the way this is presented in this manuscript implies that positive tests are a desirable outcome that "should be increased", in clinical practice this is not the case.

RESPONSE: We agree that in practice a positive or negative genetic test results informs patient care. To re-focus the reader on the effects of interventions on GC referral and completion and GT completion lines 337-338 have been edited to remove “and minimally on patients identified with hereditary cancer” to remove implying a positive test result is a desired outcome. Additionally, throughout the results the focus on interventions effects on GC referral, completion and GT completion are reported and less so on positive mutation rate detection. See tracked changed manuscript lines 344-358, 364-370,374-377,387-405. However, as you mention mutation detection rates or identifying hereditary cancer has been reported as an outcome in 56% of included studies and remains listed in Tables 3 and 4 as an outcome reported. 

COMMENT TO THE AUTHOR:

9) Line 467- should say "except" instead of "exempt"

RESPONSE:

We thank the reviewer for highlighting the need to edited this sentence. See track changed manuscript lines 476 for the correct use of the word “except”.

10. COMMENT TO THE AUTHOR:

Lines 567-571- Given the wide variety of practice settings and heterogeneity in oncology clinics around the world, I don't know that it is fair to imply that there are uncertainties about the "true effects of interventions...". While single site studies do limit generalizability, they are in fact very helpful for informing possible interventions that might be effective *at similar clinics*. For example, a clinic intervention that trains nurses to do pre-test consent and sees an increase in the proportion of ovarian cancer patients who complete genetic testing prior to starting treatment would be highly valuable for other clinics who do not have an on-site genetics clinic but do have nurses who have access to genetics training, while other clinics who have on-site "embedded" genetic counselors in their gyn clinic might not have a use for that particular intervention. In other words, heterogeneity is not necessarily a bad thing. I note that this may be more of a fact for US-based oncology clinics vs clinics in systems with universal health care where there is a more uniform approach.

RESPONSE: We thank the reviewer for this helpful and valid point regarding the positive aspects of heterogeneity in practice settings. We have edited lines 576-583 to recognise the limitations on generalisability of single site urban hospital settings and acknowledged that implementation lessons can still be garnered for similar clinic settings. We have deleted reference to the uncertainties of the true effects of the intervention as we agree this is not possible to prove. See track changed manuscript lines 576-583.

REVIEWER 2

The authors completed a SR on integrating genetic testing into routine oncology care. This SR covers a relevant research question and the authors report on key outcomes that are especially important for researchers trying to implement genomic programs. Although well written and the methods are well documented, a few minor comments are suggested below to help enhance the analysis.

1. COMMENT TO THE AUTHOR:

Abstract:

• Line 43: what was the cutoff date? 2011-?

• Line 46: define complex intervention. Quantify “increase in access to genetic counseling and testing in routine oncology practice”. How much of an increase?

RESPONSE: We thank the reviewer for their comments and have edited line 43 to reflect that studies up to May 2020 were included. A complex intervention is defined as (multiple components) now place in brackets after the word complex to clarify. An average rate of increase in access to genetic counselling and test completion has been included in lines 48-49, see tracked changed manuscript.

2. COMMENT TO THE AUTHOR:

Background:

• Line 64: specify the type of genetic testing. Somatic vs germline testing.

• Line 111: provide an example of a single unit vs complex intervention

RESPONSE: Line 64 has been updated to specify “cancer germline genetic testing (GT)…” See tracked changed manuscript line 65. 

Lines 117-124 have been included to give more detail on defining a complex intervention and examples of a single unit versus complex intervention to demonstrate the difference. 

3. COMMENT TO THE AUTHOR:

Methods

• Line 133: Although not required, it would be helpful to include timing/length of follow up as part of the inclusion/exclusion criteria.

• Line 193: double check how the tables/figures are numbered. Table 3 is mentioned here but Tables 1 and 2 were not mentioned prior to this.

• Line 201: how was the quality of RCTs assessed?

RESPONSE: Timing and length of follow up for studies with a control are reported in the Results under the quality appraisal section lines 271-274. We have not included in the inclusion criteria, as this factor was not used as an inclusion or exclusion item.

We thank the reviewer for highlighting the discrepancy in table numbering. We have deleted reference to Table 3 in the methods section line 237 and 240. Table numbering is now in order in the revised manuscript. 

There were no eligible RCT found as per inclusion criteria in the review of studies, thus no reference to quality of RCT is included in the quality apprasial section. Mainly observational and qualitative studies were included and quality assessed

4. COMMENT TO THE AUTHOR:

Results:

• Line 277: I like how the studies were mapped to outcomes and CFIR constructs. One challenge in implementing programs is knowing which outcomes and constructs to use and the optimal combination of them. These outcomes and constructs are reported individually, but what would be helpful from a practice standpoint is knowing the combination of constructs that studies implemented and seeing how that impacted outcomes. Perhaps the CFIR constructs could be included in Table 4?

RESPONSE: All of the framework mapping results (Outcomes and CFIR) are now included in Table 4 to show which outcomes mapped to the frameworks. See tracked changes in Table 4. All of these framework mapping are also included in Table in S4 Table

• It’s a bit difficult to follow all the tables. Perhaps to simplify this, a diagram outlining how the studies were mapped to Proctor and CFIR could be diagrammed and mention of which table the results are in could be referenced in the figure. Just a thought to help the reader know which table to reference for specific information.

RESPONSE:Now that all framework mapping has been included in Table 4 we think that this will help the reader to follow the framework mapping results instead of having to go to the supplemental material

Tables:

• Is there a table where the study design is summarized? This would be helpful to know in tables such as Table 4.

RESPONSE: The study design are now included at the end of Table 2 and Table 4 in the Study Quality and design column.

• Table 1: I’m still confused about what a “complex intervention type” is. Is “Education” considered complex because there were 4 implementation strategies that were involved in a study? And what about the single units? Is there a list of these somewhere?

RESPONSE: Complex intervention is defined in more detail in lines 117-124 with an example of a single unit versus multicomponent interventions. An example of a single unit intervention is from Cohen et al43 that use one component - Genetics attendance at an MDT tumour board meeting in gynaecology oncology. The remainder of the intervention studies included used complex intervention types with multiple components.

Supplemental materials:

• All titles for tables and figures should start with “Supplemental”.

RESPONSE: All tables and figures now begin with Supplemental (S), see track changes in Supplemental material

None of the named authors have a conflict of interest, financial or otherwise.

Many thanks for considering our revised manuscript for publication. We look forward to learn of the outcome

Sincerely,

Rosie O Shea, BSc, MSc

Senior Genetic Counsellor

PhD Candidate, Faculty of Medicine and Health 

University of Sydney

Sydney, Australia

Tel: +61 447733582

Email: rosie.oshea@sydney.edu.au

https://orcid.org/0000-0002-9186-1644

---

## [Decision Letter · Decision Letter 1]

6 Apr 2021

Health system interventions to integrate genetic testing in routine oncology services : a systematic review

PONE-D-20-35958R1

Dear Dr. O' Shea,

We’re pleased to inform you that your manuscript has been judged scientifically suitable for publication and will be formally accepted for publication once it meets all outstanding technical requirements.

Kind regards,

Alvaro Galli

Academic Editor

PLOS ONE

Additional Editor Comments (optional):

Reviewers' comments:

Reviewer's Responses to Questions

**Comments to the Author**

1. If the authors have adequately addressed your comments raised in a previous round of review and you feel that this manuscript is now acceptable for publication, you may indicate that here to bypass the “Comments to the Author” section, enter your conflict of interest statement in the “Confidential to Editor” section, and submit your "Accept" recommendation.

Reviewer #1: All comments have been addressed

Reviewer #2: All comments have been addressed

2. Is the manuscript technically sound, and do the data support the conclusions?

Reviewer #1: Yes

Reviewer #2: Yes

3. Has the statistical analysis been performed appropriately and rigorously? 

Reviewer #1: N/A

Reviewer #2: Yes

4. Have the authors made all data underlying the findings in their manuscript fully available?

Reviewer #1: Yes

Reviewer #2: Yes

5. Is the manuscript presented in an intelligible fashion and written in standard English?

Reviewer #1: Yes

Reviewer #2: Yes

6. Review Comments to the Author

Reviewer #1: (No Response)

Reviewer #2: The authors did a great job addressing my comments (as well as the comments from the other reviewer). No further revisions are needed at this time. This is a valuable systematic review that adds to the literature in a meaningful way.

7. PLOS authors have the option to publish the peer review history of their article (what does this mean?). If published, this will include your full peer review and any attached files.

Reviewer #1: No

Reviewer #2: No

---

## [Editor Report · Acceptance letter]

15 Apr 2021

PONE-D-20-35958R1 

Health system interventions to integrate genetic testing in routine oncology services: a systematic review 

Dear Dr. O'Shea:

I'm pleased to inform you that your manuscript has been deemed suitable for publication in PLOS ONE. Congratulations! Your manuscript is now with our production department. 

Kind regards, 

on behalf of

Dr. Alvaro Galli 

Academic Editor

PLOS ONE